# DNA replication dynamics during erythrocytic schizogony in the malaria parasites *Plasmodium falciparum* and *Plasmodium knowlesi*

**Jennifer McDonald, Catherine J. Merrick** *

Department of Pathology, University of Cambridge, Tennis Court Road, Cambridge, United Kingdom

* cjm48@cam.ac.uk

**Data Availability Statement:** All relevant data are within the manuscript and its Supporting Information files.

## Abstract

Malaria parasites are unusual, early-diverging protozoans with non-canonical cell cycles. They do not undergo binary fission, but divide primarily by schizogony. This involves the asynchronous production of multiple nuclei within the same cytoplasm, culminating in a single mass cytokinesis event. The rate and efficiency of parasite reproduction is fundamentally important to malarial disease, which tends to be severe in hosts with high parasite loads. Here, we have studied for the first time the dynamics of schizogony in two human malaria parasite species, *Plasmodium falciparum* and *Plasmodium knowlesi*. These differ in their cell-cycle length, the number of progeny produced and the genome composition, among other factors. Comparing them could therefore yield new information about the parameters and limitations of schizogony. We report that the dynamics of schizogony differ significantly between these two species, most strikingly in the gap phases between successive nuclear multiplications, which are longer in *P. falciparum* and shorter, but more heterogenous, in *P. knowlesi*. In both species, gaps become longer as schizogony progresses, whereas each period of active DNA replication grows shorter. In both species there is also extreme variability between individual cells, with some schizonts producing many more nuclei than others, and some individual nuclei arresting their DNA replication for many hours while adjacent nuclei continue to replicate. The efficiency of schizogony is probably influenced by a complex set of factors in both the parasite and its host cell.

## Author summary

Malaria parasites are unusual, early-diverging single-celled organisms. One of their atypical features is their mode of producing new cells. Most cells replicate their genome, segregate the copies into two nuclei and then split the whole cell in two: a process called binary fission. Malaria parasites, by contrast, multiply primarily by schizogony. Each cell replicates its genome several times over, asynchronously, generating many nuclei within the same cytoplasm, before splitting into many new daughter cells in a single mass event. All malaria species do this, but there are stark differences between species in how long

**Funding:** This work was supported by a European Research Council (https://erc.europa.eu/) Research grant, 'Plasmocycle' (725126) to CJM. The funders had no role in study design, data collection and analysis, decision to publish, or preparation of the manuscript.

**Competing interests:** The authors have declared that no competing interests exist.

schizogony takes and how many progeny each cell can produce. Understanding this is important because the rate of parasite reproduction is fundamentally important to malarial disease: humans who have a high burden of parasites in their blood are most likely to suffer severe malaria. Here we compare the process of schizogony in two different species, by developing cell lines in which we can follow *de novo* DNA replication at high spatial and temporal resolution, at both whole-cell and single-molecule levels. We establish the dynamics of schizogony, highlighting similarities and differences between two species and setting parameters for future studies of interventions that could interfere with parasite reproduction.

## Introduction

The genus of malaria parasites *Plasmodium* lies in an early-diverging protozoan lineage with many unusual features. One such feature is the type of cell cycle that this parasite pursues [1]. *Plasmodium* does not divide by binary fission–the well-studied process that occurs in almost all model systems from yeasts to human cells. Instead, new cells are produced primarily by schizogony, a unique process involving asynchronous DNA replication in multiple nuclei within the same cytoplasm, prior to a mass cytokinesis event producing many–and not necessarily $2^n$–daughter cells called merozoites [2].

Schizogony occurs first in hepatocytes and then repeatedly in erythrocytes, causing all the pathology of malaria. The process is clinically important because the rate of growth in erythrocytes determines parasitaemia, which often correlates with disease severity [3, 4]. Parasitaemia in human malaria can vary greatly, partly because the human-infective *Plasmodium* species (*P. falciparum*, *P. vivax*, *P. malariae*, *P. vivax*, *P. knowlesi* and *P. ovale wallikeri* & *curtsii*) differ in their preferences for human erythrocytes. For example, *P. vivax* is restricted to invading scarce reticulocytes, *P. malariae* primarily invades older erythrocytes, and *P. falciparum* invades cells of all ages and can therefore reach high parasitaemias as extreme as 40% [5].

There are also other inter-species differences in the process of schizogony. Its duration varies from ~72 hours in *P. malariae* to ~24 hours in *P. knowlesi* and the number of merozoites produced per schizont varies from as many as ~30 in *P. falciparum* to ~15 in *P. knowlesi*. Exactly what limits merozoite numbers to a characteristic range for each species is unknown, but it is evidently not the fundamental size of the human erythrocyte, nor the duration of schizogony. Moving from the cell-biological to the molecular level, the composition of *Plasmodium* genomes also varies considerably between species. *P. falciparum* has an extremely biased genome at ~81% A/T [6] whereas *P. vivax* and *P. knowlesi* have A/T contents of only 60 and 61% [7, 8]. The reasons for these differences, and their potential influence upon the speed and fidelity of DNA replication, are unknown.

To better understand the dynamics of *s*chizogony, we have compared the process in the only two human-infective species that are amenable to culture, *P. falciparum* and *P. knowlesi*. These differ in parameters such as genome content, cell cycle period and merozoite number. Little has been published on this subject in *P. knowlesi* but we have previously explored DNA replication dynamics in *P. falciparum* [9]. To do this we created a parasite line that allows *de novo* DNA replication to be followed at high resolution via the incorporation of pulse-labels of modified nucleotides [10]. Thus, we defined DNA replication dynamics at the single-molecule level [9] and, together with prior work at the cellular level [2, 11], this provided an overall picture of *P. falciparum* erythrocytic schizogony. Here, we have added considerable detail to that picture and used the same DNA-replication-labelling technology to compare *P. knowlesi*–a

species that completes schizogony twice as fast, makes fewer merozoites, and has a more balanced genome composition.

To summarise what is already known about *P. falciparum* schizogony, the first round of DNA replication begins more than halfway through the cell cycle (which is theoretically 48h, actually ~42-50h in different strains and culture systems [2, 12]). Four or five rounds of nuclear multiplication occur and there is karyokinesis at each round [2, 11], without cytokinesis or conventional G1/G2 phases [2, 11, 13]. Duplication of the centriolar plaque (the *Plasmodium* centriole-equivalent) has been proposed to initiate each round of DNA replication [2]. Asynchrony, which is clearly seen in the early rounds, has been linked to partitioning of centriolar plaques, with the nucleus that receives the more mature plaque commencing its next DNA replication first [2]. Without the benefit of DNA replication labelling, prior work could not define the later replicative rounds precisely, as nuclei become very crowded inside the erythrocyte, but the final round was suggested to be synchronous [12, 14]. The degree of synchrony has since been disputed, most recently via high-resolution electron-microscopy data [2, 15]. Indeed, it is unclear how a final synchronous round would be coordinated if centriolar plaques of differing maturity do drive the rate of DNA replication. Other unanswered questions include: is there a limit on the number of nuclei that can replicate DNA at once, perhaps via a limiting factor, and how is this overcome if the final round is synchronous? Do all successive rounds of nuclear multiplication take the same time? Are there consistent gaps between replicative rounds (G1 or G2 equivalents), or do some nuclei 'stall'–adding to asynchrony and suggesting a DNA replication checkpoint?

At the single-molecule level, our recent work defined the average speed of replication forks and spacing of replication origins in *P. falciparum* schizogony at 1.2kb/min and 65kb between origins–broadly similar to parameters in other eukaryotic cells. These averages changed by ~30% over the course of schizogony, with the fastest fork movement and most widely-spaced origins occurring early on [9]. The opposite pattern occurs in human cells, where DNA replication speed is usually limited by cellular nucleotide pools and becomes fastest towards the end of S-phase as nucleotide production peaks [16]. By contrast, in *P. falciparum*, we speculated that the nucleotide pool could become increasingly limiting as more nuclei replicate their DNA at once, and/or that DNA condensation or accumulated DNA damage might limit the pace of replication [9]. In *P. knowlesi*, fewer nuclei are packed into each cell and the genome composition is different from that of *P. falciparum*, so the parameters of DNA replication could conceivably be very different. Accordingly, we have now measured these parameters at both the cellular and single-molecule levels in both species.

## Results

### Timecourse experiments across S-phase in *P. falciparum* and *P. knowlesi*

To follow DNA replication dynamics throughout schizogony in both *P. falciparum* and *P. knowlesi*, we first transfected *P. knowlesi* with a thymidine-kinase-expressing plasmid, as previously performed in *P. falciparum* [10]. We confirmed that this permitted the labelling of DNA replication with modified nucleotides (bromo/chloro/iodo/ethyl-deoxyuridine (BrdU/CldU/IdU/EdU)), and that parasite behaviour in terms of fitness and cell-cycle timing was not overtly disturbed (S1 Fig).

We then conducted time-course experiments with highly synchronised cultures of both lines. Synchronisation was achieved by purifying late-stage schizonts, allowing 1h of reinvasion, then purifying the resultant ring-stages to yield a 1h window around the point of invasion (termed '0 hpi'). Aliquots of parasites were pulse-labelled with EdU for 30 minutes every hour throughout the period of DNA replication, then immediately fixed and prepared for

microscopy. All the nuclear masses in each cell were detected with a DNA stain, and all those that were *actively replicating* were detected via EdU click chemistry. (A nuclear mass refers to a distinguishable mass of DNA that may or may not be a discrete nucleus–i.e. fully enveloped in a nuclear membrane–since we were unable to stain this membrane.) Centrin foci were also detected by immunofluorescence [2], to quantify the centriolar plaques associated with each nuclear mass. Fig 1A shows a schematic of the experimental setup. In both species, additional experiments were then conducted, using two distinguishable pulses of modified nucleotides at increasing time intervals (Fig 1B and 1C). This allowed the detection not only of nuclei actively replicating their DNA at time x, but also of nuclei that had been replicating at time x-y. Thus, we could determine the length of each replicative round and of the gaps between rounds.

## The length of S-phase is proportional to the overall length of erythrocytic schizogony

To measure the overall parameters of S-phase, we first counted all the nuclear masses, and all those that were actively replicating DNA, in 100 cells at each timepoint listed in Fig 1A (S2 Fig). *P. falciparum* schizonts produced 5–28 nuclei while *P. knowlesi* produced 3–16 (Fig 2A and 2B). Median numbers of nuclei at the most mature timepoint were 11 for *P. falciparum* and 6 for *P. knowlesi*, but the numbers produced in both species were highly variable: amongst populations of late-stage schizonts, maximum numbers of merozoites were 28 and 16, but rare cells were still visible with only 4 or 2 nuclei respectively (Fig 2A and 2B). This could be because a small minority of cells were very delayed in starting S-phase, or because a minority underwent fewer/slower rounds of nuclear multiplication. Favouring the latter explanation, the onset of S-phase for the majority of cells was quite synchronous (Fig 2C and 2D), particularly in *P. knowlesi*, where ~60% of cells commenced DNA replication within a single hour at 22-23hpi. Median numbers of nuclear masses counted at the latest timepoints were markedly lower than in previous studies that counted only mature schizonts, arrested or visually selected at the point of egress [12, 17]. This is because we included all cells, counted agnostically at each timepoint, rather than only segmented schizonts, and therefore captured the full diversity of the population, including cells that may have experienced S-phase delays and/or fewer rounds of nuclear multiplication.

The overall length of S-phase was estimated as the time between the first appearance of cells with actively replicating DNA and the time when the number of individual nuclear masses stopped increasing. In *P. falciparum*, cells with replicating DNA began to appear at 31-32hpi and cells with more than 1 nuclear mass (having completed the first S-phase) began to appear at 32hpi (Figs 2A and 2C, and S3). Numbers of nuclear masses stopped increasing at 46-47hpi, making 15 hours the total S-phase period across the population (importantly, this may not apply to every cell within that population, and could also be slightly influenced by strain-to-strain variation in strains other than 3D7). In *P. knowlesi*, DNA replication started at 22hpi, 2n cells began to appear at 23hpi, and numbers of nuclear masses stopped increasing at 31-32hpi (coinciding with a sharp drop in EdU labelling from 31hpi), i.e. an S-phase period of only 9h (Figs 2B and 2D, and S3). Notably, this *P. knowlesi* timecourse was followed until almost all the schizonts had burst at 33hpi, confirming a lag of just 2h between the cessation of DNA replication and the completion of reinvasion. In *P. falciparum*, this first timecourse was not followed through to the end of reinvasion but a replicate experiment (S3 Fig), as well as the subsequent EdU/BrdU labelled timecourse would confirm that *P. falciparum* showed substantial reinvasion by 48hpi (S3 Fig), and so has a similar window of ~2h between the end of S-phase and reinvasion. Within this window, daughter merozoites must be fully assembled and cytokinesis must occur.

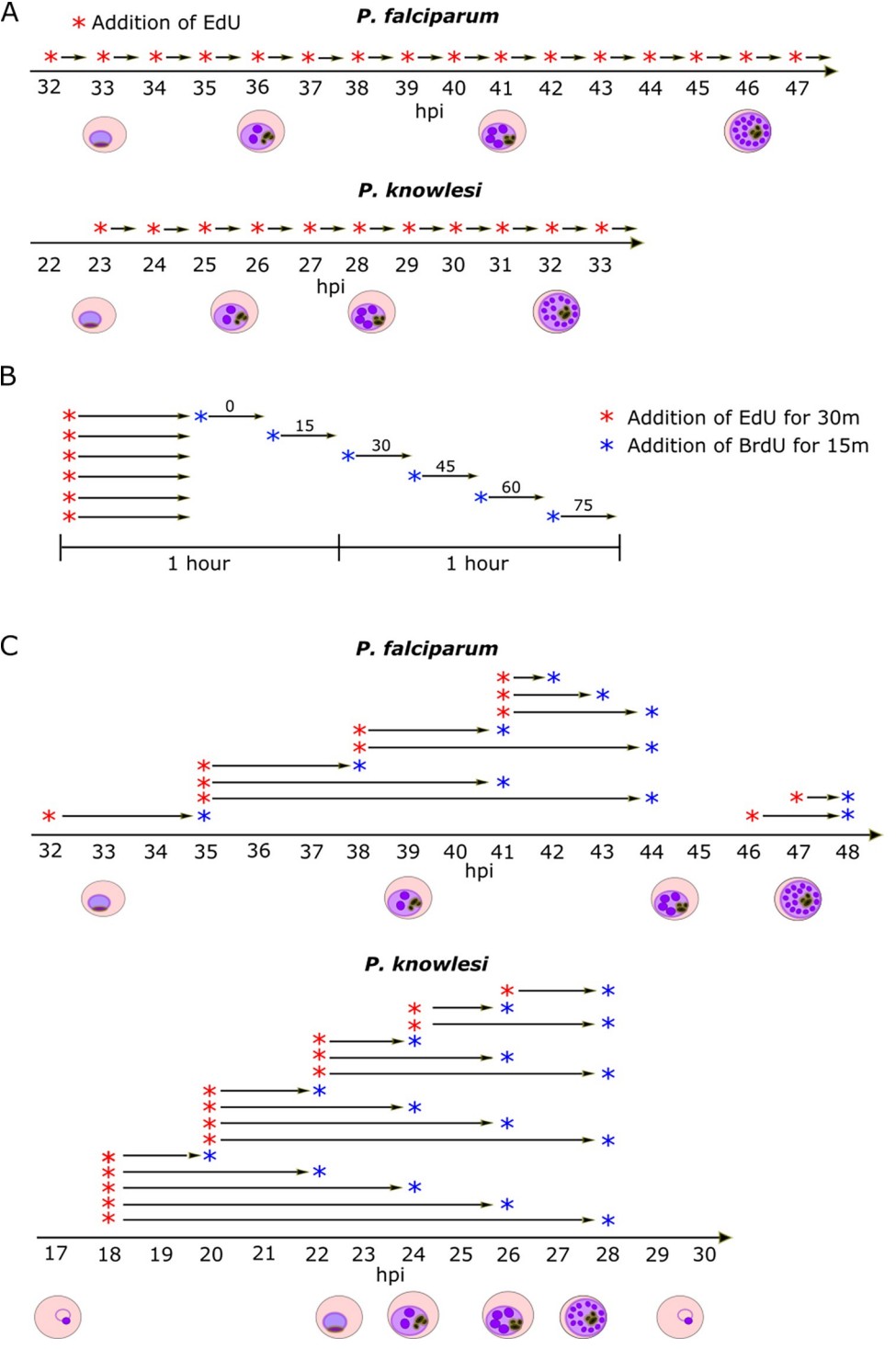

**Fig 1. Schematics of experimental setup.** A: Schematics of timecourses using single EdU pulse labels. B: Schematic of experiments using sequential pulses of two nucleotides (EdU and BrdU) to determine the length of each replicative round. C: Schematics of timecourses using 2-nucleotide (EdU and BrdU) interval pulse labels.

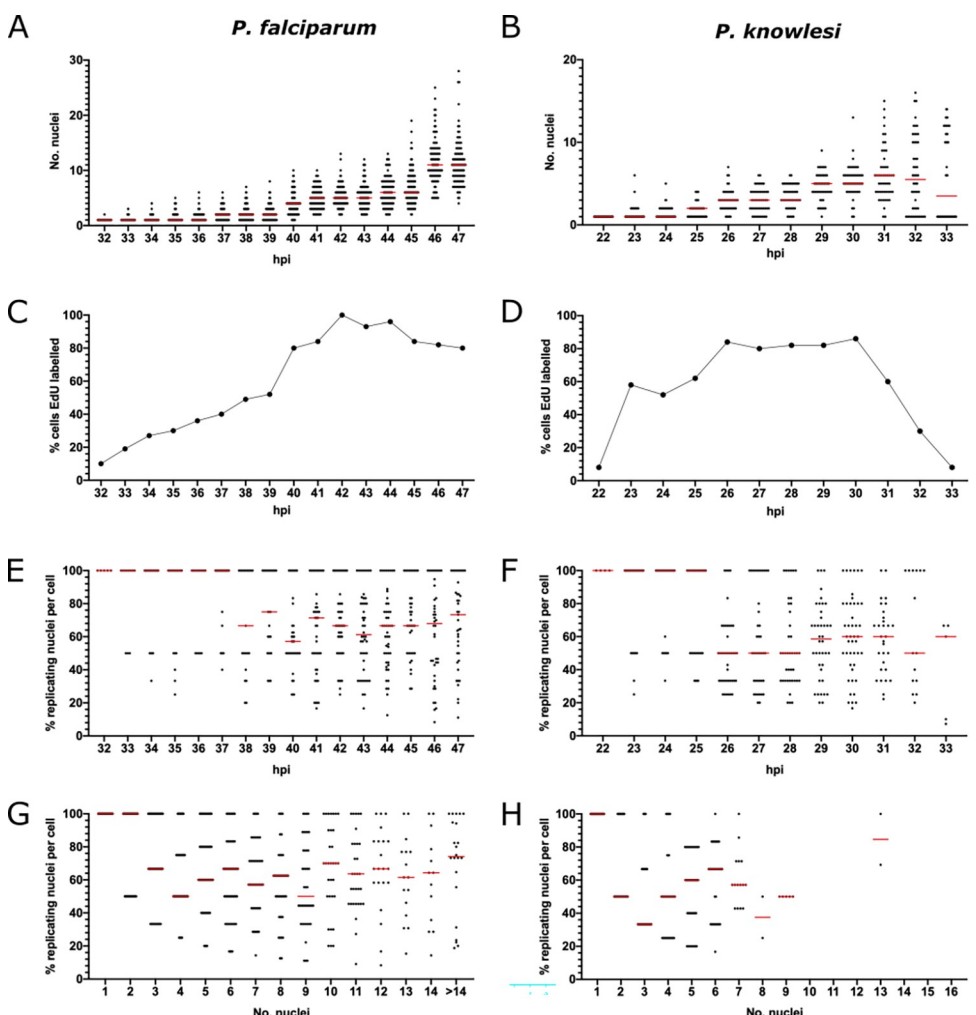

**Fig 2. Length and dynamics of S-phase in *P. falciparum* and *P. knowlesi*.** A, B: Scatter graphs of nuclear numbers at each hour in *P. falciparum* (A) or *P. knowlesi* (B), n = 100 cells, medians are shown in red. Biological replicate experiments are shown in S3 Fig. C, D: Percentage of 100 cells showing some EdU labelling at each hour in *P. falciparum* (C) and *P. knowlesi* (D). E, F: Percentage of replicating nuclear masses per cell at each timepoint in *P. falciparum* (E) and *P. knowlesi* (F), n = 100, medians are shown in red. G, H: Percentage of replicating nuclear masses per cell, re-plotted by number of such masses, in *P. falciparum* (G) and *P. knowlesi* (H), n = 100, medians are shown in red.

These data showed that the proportion of the cell cycle occupied by S-phase was very similar in both species, at ~30% (15 of 48h, or 9 of 31h) and that the lag period between nuclear multiplication and reinvasion was similarly short in both species. All parameters shown in Fig 2 were broadly reproducible in separate timecourses conducted as full biological replicates in different blood batches on different days (S3 Fig). Variation in batches of host erythrocytes can influence *Plasmodium* cell cycles, as documented in previous work [12, 17]. In our replicates, S-phase in *P. falciparum* proceeded with similar timing to the previous timecourse, between ~32-47hpi, but a slightly higher peak in median numbers of nuclear masses was counted at 47hpi. The replicate timecourse in *P. knowlesi* produced similar numbers of nuclear masses but was set forward by several hours (and likewise in the subsequent EdU/BrdU labelled timecourse). In our hands, the cycle time of *P. knowlesi* is particularly sensitive to different batches of human erythrocytes: ~28h is common in culture and the initial timecourse shown

in Fig 2 was therefore unusually long, possibly due to the age of the erythrocytes in that blood batch.

## There is no restriction on the number of nuclei that can replicate at once

DNA replication in schizogony is clearly a stochastic, asynchronous process and different nuclei within the same cell can simultaneously experience either DNA replication or resting 'gap' phases [2, 11]. We therefore examined whether there is a limit on the number of nuclei that can replicate simultaneously. If, for example, the pool of nucleotides or replication proteins is limiting, we should observe a limit on the number of nuclei per cell that can replicate their DNA within a 30-minute period. In fact, throughout schizogony this was not true. In the earliest stages of S-phase, the percentage of labelled nuclear masses was either 50 or 100%, since cells had only 1 or 2 nuclei. Subsequently, however, cells appeared with anything from 20% to 100% of their distinguishable nuclear masses actively replicating DNA. The median percentage at most timepoints was 60–70% in *P. falciparum* and 50–60% in *P. knowlesi* (Fig 2E and 2F).

To obtain a clearer view of whether the number of nuclei replicating their DNA at once depended on the number of nuclei existing in a schizont, we re-plotted the data according to number of existing nuclear masses rather than hpi (Fig 2G and 2H). This confirmed the conclusion that a schizont with any number of nuclei could replicate DNA in any proportion of those nuclei within a 30-minute window. Notably, in *P. knowlesi* DNA replication was rarely observed after the stage of 8–9 nuclear masses (Fig 2H), consistent with this species reaching a maximum of 16 nuclei, and with the final round of DNA replication being quite, but not entirely, synchronous. By contrast, numbers of nuclear masses in *P. falciparum* schizonts were highly variable and cells with 15 or more nuclei were therefore binned together (Fig 2G), but there was no single number of nuclei (even above 15) at which DNA replication ceased, indicating greater variability and asynchrony in *P. falciparum*.

## Patterns of DNA replication within nuclei vary from early to late schizogony

In both *P. falciparum* and *P. knowlesi*, it was clear that the sub-nuclear distribution of actively-replicating DNA during the 30-minute pulse-label varied in different nuclei (Fig 3B). The commonest pattern was a completely labelled nuclear mass with signal detected throughout the DAPI-stained area, showing that DNA in all areas had replicated in the previous 30 minutes. Amongst these, and particularly notable in *P. falciparum*, was a subset with homogenous but fainter staining, suggesting that DNA had replicated much less rapidly than in the common pattern: this appeared mostly in late-stage cells completing their presumptive final round of nuclear multiplication (Fig 3C, see *P. falciparum* at 44-45hpi, *P. knowlesi* at 30hpi, and quantification in Fig 3D and 3E). Other patterns included nuclear masses with only a partial area showing DNA replication, and nuclear masses with discrete foci of DNA replication: these appeared at low frequencies throughout both timecourses (Fig 3D).

To quantify these observations, we categorised DNA replication patterns in the nuclei of 100 cells at each timepoint (Fig 3D). We also quantified the amount of de novo DNA replication occurring within the pulse-label via the density of EdU staining per nuclear mass (Fig 3E), confirming the late-stage dominance of the 'homogenous faint' DNA replication pattern (individual patterns are shown as separate graphs in S4 Fig). Marked differences appeared between *P. falciparum* and *P. knowlesi*. Firstly, nuclear masses with the 'partial' DNA replication patterns made up 20–30% of the total in *P. falciparum* at all timepoints, but fewer than 10% of the total in *P. knowlesi* at all times until the final timepoint (Fig 3D). Secondly, the majority of

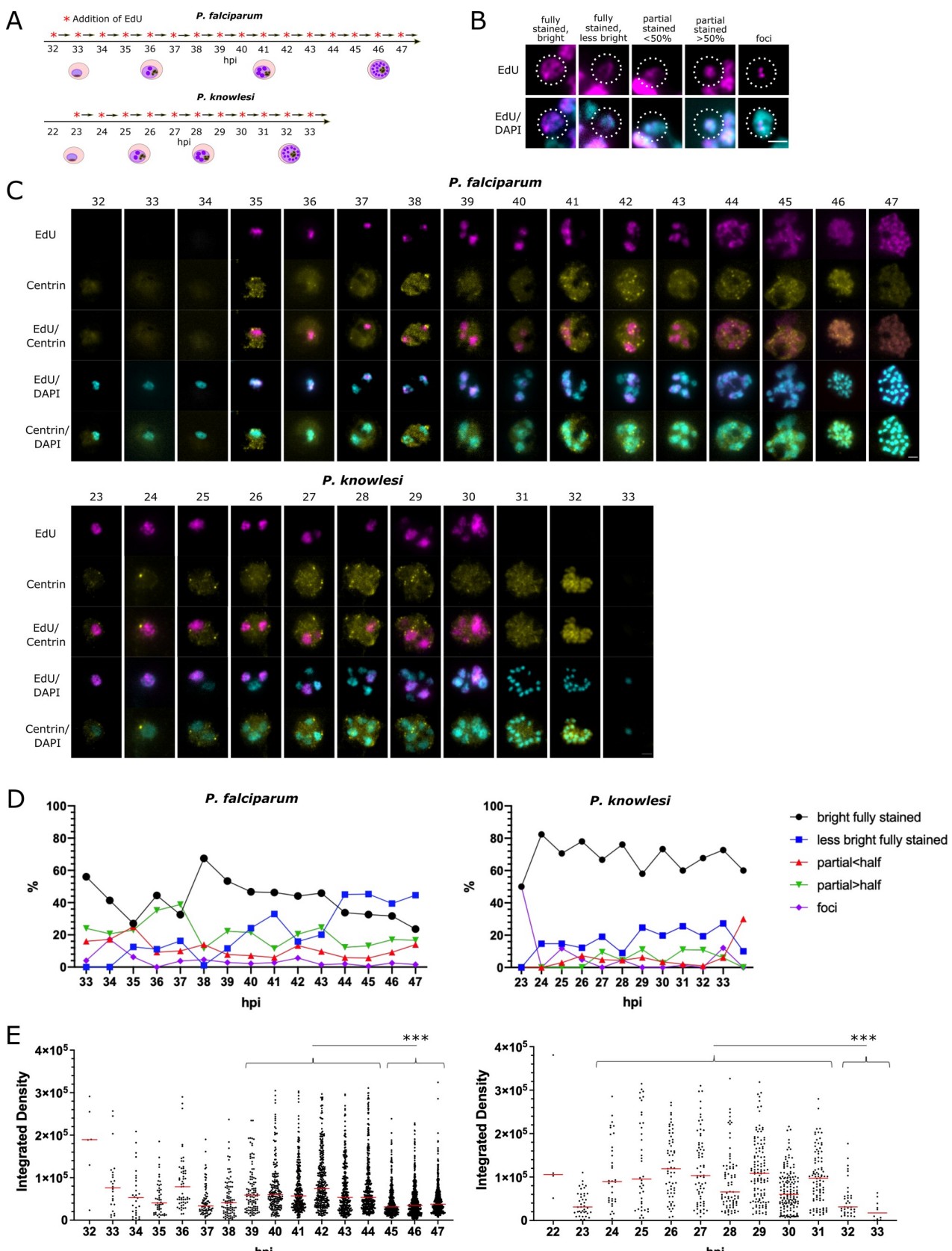

**Fig 3. Patterns of DNA replication within nuclei in *P. falciparum* and *P. knowlesi*.** A: Schematic of the timecourses shown in this figure. B: Examples of 5 distinct patterns of EdU incorporation seen in replicating nuclei (*P. falciparum* are shown, representative of patterns in both species; dotted line highlights the relevant nuclear mass). Scale bar all panels 1μm. C: Representative examples of cells at each hour across the timecourse for *P. falciparum* and *P. knowlesi*. 46-47hpi in *P. falciparum* and 31-32hpi in *P. knowlesi* show segmented schizonts; 33hpi in *P. knowlesi* shows a reinvaded ring. Scale bar all panels 2μm. D: Percentages of each distinct pattern seen at each point across the timecourse in *P. falciparum* and *P. knowlesi*. E: Intensity of EdU staining (calculated as 'integrated density' across each nuclear mass, i.e. the sum of values for fluorescent signal in all pixels in each mass) seen at each timepoint in *P. falciparum* and *P. knowlesi*. A significant drop in EdU staining intensity is seen at the end of the timecourse in both species (\*\*\*, p = 0.001, ANOVA). In *P. falciparum*, this correlates clearly with the rise at 44-47hpi in the pattern qualitatively categorised as 'less bright' in (C). Intensities in nuclear masses with each pattern of staining are shown separately in S4 Fig. Nuclei that had entirely finished replicating and showed no detectable signal were excluded.

nuclear masses in *P. falciparum* showed the faint staining pattern at 44-47hpi, whereas in *P. knowlesi* this pattern increased slightly but never became dominant, appearing in only ~20% of nuclear masses (Fig 3D and 3E).

## Centriolar plaques are dynamic throughout schizogony

Several previous studies have highlighted the importance of the centrosome equivalent–the centriolar plaque–in controlling nuclear multiplication [2, 18] (S5A Fig). We therefore quantified whether or not nuclei in each phase of DNA replication were associated with detectable centriolar plaques, using foci of centrin as a marker. Importantly, these experiments relied upon a commercial anti-centrin antibody, the sensitivity of which was called into question in recent work (published after this study was conducted) [18]. Using this marker, centrin foci were not always detectable (S5B and S2 Figs): overall, ~50% of nuclear masses had detectable centrin foci, varying from <30% in the earliest and latest timepoints to >60% at the midpoint of schizogony (37-42hpi in *P. falciparum* and 25-28hpi in *P. knowlesi* (S5C Fig)). Only a minority of nuclear masses, ~25% at most timepoints, showed centrin foci as well as active DNA replication, while the majority showed either DNA replication with no centrin, or centrin with no DNA replication (S5D Fig).

The data were broken down into distinct types of centrin labelling: a single focus, two separate but adjacent foci, and two foci on opposite sides of the nuclear mass (S2 Fig). These are thought to represent sequential events in preparation for karyokinesis (S5A Fig). Across 100 cells, in both *P. falciparum* and *P. knowlesi*, duplicated centrin foci were relatively rare, particularly in the opposite configuration, showing that this phase is very brief (S5C Fig). About half of all nuclear masses with two foci showed DNA replication within the past 30 minutes, while the other half did not (S5D Fig). Finally, nuclear masses with two centrin foci were never seen past the stage of 8–9 such masses in *P. knowlesi* (S5E and S5F Fig). This division was less sharp in *P. falciparum*, but the appearance of centrin foci did drop off in cells with more than ~10 nuclear masses. In both species, it therefore appears than centriolar plaques are disassembled after the final round of DNA replication and karyokinesis. Beyond this, it was not possible to discern a clear relationship between the pattern of intranuclear DNA replication (full, partial, discrete foci, etc.) and the pattern of centrin foci (S5E and S5F Fig).

## Replicative rounds become faster across the course of schizogony

The foregoing experiments could not measure the time taken by each round of nuclear multiplication, because they provided only snapshots at successive timepoints. Therefore, we elaborated the protocol to use two successive, distinguishable pulse-labels (Fig 4A). Cultures were labelled with EdU for 30 minutes (as before), then with BrdU at 15-minute intervals up to 120 minutes in total. In a 1-n trophozoite, a single nucleus bearing both labels (Fig 4B) must have been replicating DNA from at least the end of the first pulse to the start of the second, thus measuring the potential length of S-phase. Accordingly, as shown in Fig 4A, when 30 minutes

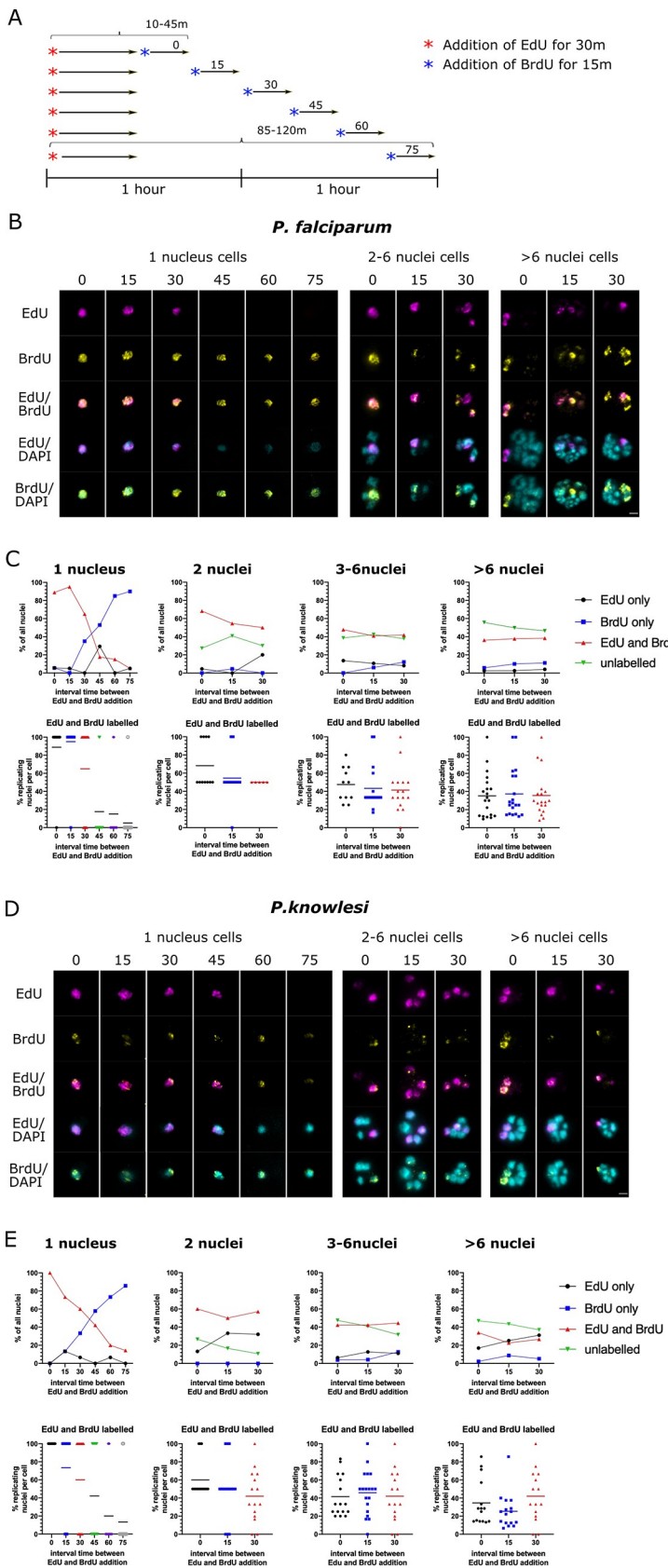

**Fig 4. Replicative rounds become faster across the course of schizogony.** A: Schematic of the timecourses shown in this figure. (Biological replicate experiments are shown in S7 Fig). B: Representative examples of *P. falciparum* cells labelled with EdU for 30 minutes and then a 15-minute pulse of BrdU at increasing intervals. Scale bar all panels 2μm. C: Graphs showing the percentage of all nuclear masses that labelled with EdU and BrdU, firstly in mono-nuclear cells and then for 2-n, 3-6n and >6-n cells (n = 20). Scatter plots display the same data broken down per-cell, with means shown for each dataset. D: Representative examples of *P. knowlesi* cells labelled with EdU for 30 minutes and then a 15-minute pulse of BrdU at increasing intervals. Scale bar all panels 2μm. E: Graphs as in (C) showing the percentage of nuclear masses labelled with EdU and BrdU in *P. knowlesi* cells of increasing ploidy (n = 20 per category).

of EdU label was directly followed by 15 minutes of BrdU, S-phase could be ≥45 minutes, but minimally ~10 minutes (requiring 5 active minutes in each pulse for replicating DNA to be detected with confidence (S6 Fig)). Similarly, with a gap of 75 minutes separating the labels, S-phase must be at least 85 minutes. Fig 4C quantifies this experiment for *P. falciparum*, showing that at least 70% of nuclear multiplications in 1-n cells took 40–75 minutes, but only ~15% took 55–90 or 70–105, and almost none took >85 minutes (Fig 4C). (Notably, at the 45-minute interval, 35% of nuclei had completed DNA replication within the first pulse but did not label at all with the second, i.e. they paused for at least an hour before karyokinesis. This percentage then dropped, suggesting that very few nuclei paused for >75 minutes before dividing and starting a second round.)

In multinucleate cells, the situation is more complex because a nuclear mass could potentially have completed its DNA replication during the first label, divided in the intervening period, and then picked up the second label in a subsequent replicative round. We assumed that if most rounds of genome replication took 40–75 minutes, this would be unlikely when the interval between labels was ≤30 minutes, and therefore counted such double-labelled nuclear masses as being in the same replicative round. As ploidy increased there was a clear trend towards fewer nuclear masses being double-labelled at increasing time intervals (Fig 4C, compare 2 nuclei with <6 nuclei). Therefore, rounds became progressively faster as ploidy increased.

In *P. knowlesi* (Fig 4D), the first replicative round took markedly longer, with ~40% of nuclei taking 55–90 minutes, compared to only 15% of the equivalent nuclei in *P. falciparum*. In fact, ~15% of nuclei took as long as 85–120 minutes (Fig 4E). This marked difference was replicated in separate cultures of *P. falciparum* and *P. knowlesi* (S7 Fig). After this, however, subsequent replicative rounds seemed to speed up more than they did in *P. falciparum*. In *P. knowlesi* with >6 nuclear masses, only ~25% of those replicated for ≥25 minutes (~25% of nuclear masses showed EdU+BrdU after the 15min time interval), whereas in *P. falciparum* with >6 nuclear masses, ~40% were still taking at least 25 minutes to complete a replicative round. In fact, in *P. knowlesi* schizonts with >6 nuclear masses, some of them probably entered the next round within 40 minutes (see Fig 4E, 30min interval).

## Gaps between replicative rounds become longer across the course of schizogony

Finally, the interval-labelling protocol shown in Fig 4A was extended over longer periods, with intervals of up to 10h (Fig 5A). This was designed to assess the gaps between rounds of nuclear multiplication (and also to serve as a further replicate, confirming the reproducibility of the initial datasets). Examples of the resultant labelling patterns are shown in Fig 5B. These were quantified in terms of the percentage of nuclear masses labelled with either one or both nucleotides across 50 cells (Fig 5C).

The data indicated that a) gap phases are relatively long, b) they are longer in *P. falciparum* than *P. knowlesi*, and c) they become longer in both species over the course of S-phase. In *P.*

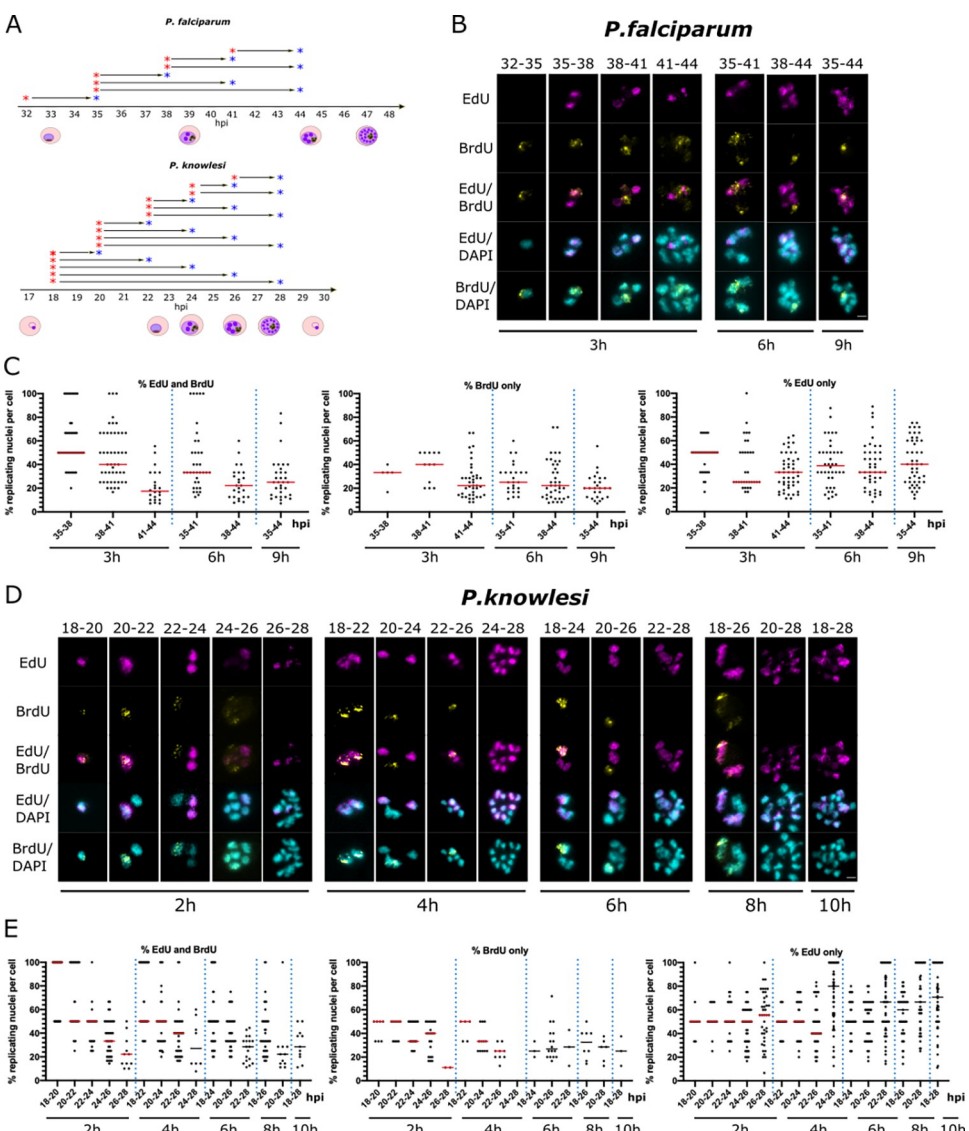

**Fig 5. Gap phases lengthen across the course of schizogony.** A: Schematic of the timecourses shown in this figure. B, D: Representative examples of cells across double-labelled timecourses for *P. falciparum* (B) and *P. knowlesi* (D). *P. falciparum* was labelled at 3h intervals, up a maximum of 9h (35-44hpi); *P. knowlesi* at 2h intervals, up a maximum of 10h (18-28hpi). Scale bars all panels 2μm. C,E: Percentages of nuclear masses per cell labelled with EdU and BrdU, EdU alone, or BrdU alone throughout each timecourse. Dotted lines separate the datasets from different time-intervals. Medians are shown in red.

*falciparum*, when sequential labels were separated by 2h (41–43 and 46-48hpi) only ~20% of nuclear masses were double-labelled, showing that relatively few nuclei commenced a second round within 2h (S8 Fig). With 3h intervals, a greater proportion of nuclei re-entered S-phase: ~50% of nuclear masses became double-labelled between 35-38hpi, but this diminished to ~40% and 20% at 38–41 and 41-44hpi (Fig 5C). Therefore, gaps increased over the course of S-phase (and/or progressively more nuclei did not re-enter DNA replication at all), even as the time spent by each nucleus actively replicating its DNA became shorter as schizogony progressed (Fig 4). By contrast, in *P. knowlesi*, gap phases were generally shorter because markedly

more nuclear masses became double-labelled within 2h: ~50% of all nuclear masses in early stages, diminishing again to ~35% and 20% at the final 2h intervals (Fig 5E).

## Individual nuclei commonly arrest for long periods during schizogony

To address the question of heterogeneity in gaps–i.e. whether nuclei can arrest for long periods, or even permanently, while others in the same cell continue to replicate their DNA, we labelled cells for 30 minutes with EdU, followed by a continuous 5h BrdU label. Nuclei showing EdU but no BrdU must have arrested for at least 5h after a period of active DNA replication: markedly longer than the common gap period of 2-3h suggested by Fig 5. Such nuclei did indeed appear often, and not exclusively in mature cells where many nuclei could have finished multiplying already (Fig 6A). In relatively young *P. falciparum* cells with ≤6 nuclear masses, ~10% of them were arrested for at least 5h (Fig 6B). In fact, this proportion fell to ~5% in >6-nucleus schizonts, but the corresponding proportion of entirely unlabelled nuclei rose (i.e. those that had ceased to replicate DNA even before the EdU pulse). In most individual cells, the number of arrested (EdU-only) nuclei was relatively low, 1–2 per cell, but rare cells were observed with more than half of their nuclei arrested for at least 5h (Fig 6C).

This phenomenon was more prominent and occurred more at earlier times in *P. knowlesi* (Fig 6D–6F). Even in relatively young schizonts, ~15% of nuclear masses were unlabelled–i.e. they had ceased to replicate DNA before the EdU pulse (see Fig 6E: ~15% of nuclei were arrested in cells with 2–6 nuclear masses, and similarly even in cells with 2–3 nuclear masses (S9 Fig)). By contrast, far fewer nuclei were labelled with BrdU only, meaning that they were unlikely to have paused throughout the EdU pulse and then commenced DNA replication again later on–which was a very common observation in *P. falciparum*. Instead, nuclei in *P. knowlesi* could either arrest long-term very early in schizogony, or go on to multiply several times with only short pauses between replicative rounds. This supports the observation from Fig 5 that the gaps between replicative rounds were shorter in *P. knowlesi* than *P. falciparum*.

## Single-molecule DNA replication dynamics do not differ significantly between *P. falciparum* and *P. knowlesi*

We previously measured the single-molecule dynamics of DNA replication in *P. falciparum*, determining that replication fork speed was ~1.2kb/min and the spacing between adjacent replication origins was ~65kb [9]. Since the genome composition of *P. knowlesi* is very different, we sought to establish whether these parameters would differ in a less A/T-biased *Plasmodium* genome.

Our previous work employed the gold-standard DNA combing technique [19, 20], which linearises single DNA fibres on glass slides at a uniform stretching factor of 2kb/μm. Cells are first pulse-labelled with two modified nucleosides, CldU and IdU, to generate DNA fibres in which the speed and directionality of replication forks can be measured (Fig 7A) [21]. Unfortunately this technique has recently become unavailable due to the discontinuation of the only antibody that reliably distinguishes CldU and IdU (previously used by all researchers in this field). The alternative double-label, EdU/BrdU, as used here in Figs 2–6, proved incompatible with DNA combing because EdU click chemistry gave poor results on the sialinized glass coverslips specifically required for DNA combing. Therefore, we reverted to the simpler technique of spreading DNA fibres from crude cell lysates on non-sialinized glass slides [22, 23]. This permitted EdU/BrdU labelling (Fig 7B), albeit at the expense of a uniform DNA stretching factor and the loss of very long fibres. DNA spreading tends to generate bundled fibres with only short lengths of single-stranded DNA, particularly when using parasite cells which have haemozoin debris in the cell lysate.

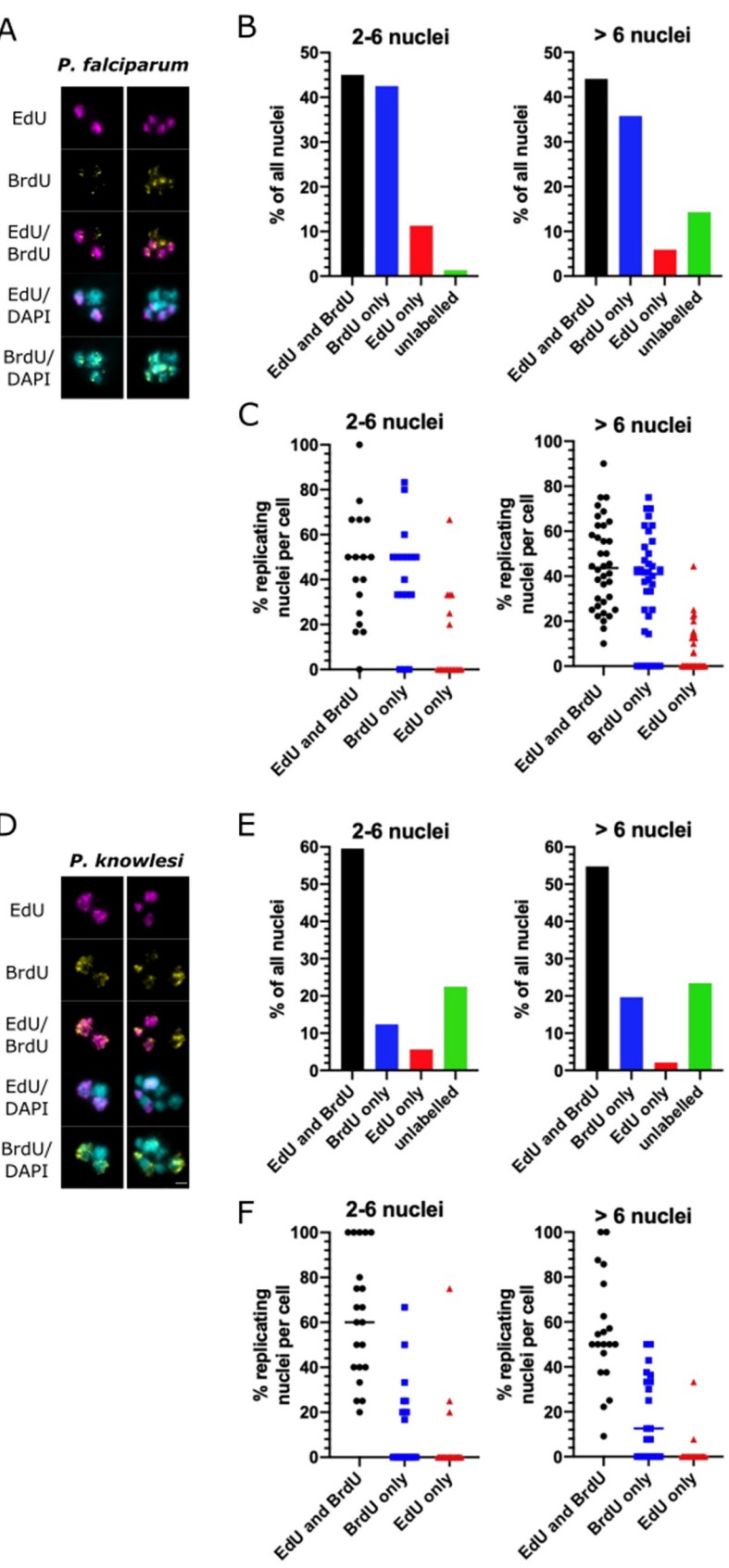

**Fig 6. Long-term arrest of individual nuclei is common during schizogony.** A: Representative examples of *P. falciparum* cells double-labelled with EdU for 30 minutes and then BrdU for 5h. Scale bar all panels 2μm. B: Percentages of all nuclear masses labelled with EdU and BrdU, EdU alone, BrdU alone, or neither label (n = 20 cells, stratified into cells with up to 6 nuclear masses or more than 6). C: Scatter plots showing percentages of nuclear masses per cell with the various labelling patterns. Medians are shown for each dataset. D: Representative examples of *P. knowlesi* cells double-labelled with EdU for 30 minutes and then BrdU for 5h. Scale bar all panels 2μm. E: Percentages of all nuclear masses labelled with EdU and BrdU, EdU alone, BrdU alone, or neither label (n = 20 cells, stratified into cells with up to 6 nuclear masses or more than 6). F: Scatter plots showing percentages of nuclear masses per cell with the various labelling patterns. Medians are shown for each dataset.

Fig 7C shows that replication fork speeds were not significantly different in the two species, although they trended faster in *P. knowlesi*. Replication origin spacing was identical in the two species (Fig 7D). Using the published DNA stretching factor of 2.59kb/μm, calculated historically on DNA spreads from human cells [23], replication fork speeds could be calculated at 0.6–0.7kb/min, and inter-origin spacing at ~30kb, but these figures are lower than our earlier measurements on combed DNA fibres from *P. falciparum* [9], and are probably less accurate because the DNA spreading method is less controlled. DNA in haemozoin-rich *Plasmodium* lysates may have a fundamentally shorter stretching factor than DNA in human cell lysates, due to, e.g., different viscosity, and measurements from DNA spreads are also skewed to include only the shorter tracts in a population, because longer tracts are more likely to enter a DNA bundle and thus be excluded–a problem that does not arise on long, single, combed DNA fibres. Nevertheless, our measurements are comparable within the conditions used here. The clear result was that DNA replication parameters did not differ significantly in *P. falciparum* versus *P. knowlesi*, although there was a trend towards faster replication fork speeds in *P. knowlesi*. This trend is corroborated by the observation that in a whole-cell context, briefer pulse-labels could be detected in *P. knowlesi* than in *P. falciparum* (S6 Fig).

## Discussion

This study is the first detailed analysis of DNA replication dynamics throughout schizogony in two different malaria parasites, *P. falciparum* and *P. knowlesi*. It reveals some key differences between the two species, such as different periods of active DNA replication and shorter gap phases between successive nuclear multiplications in *P. knowlesi* than in *P. falciparum*. There were also similarities: for example, S-phase occupied a similar proportion of the differing cell-cycle lengths in both species.

S-phase occupied ~30% of each cell cycle, i.e. 15h in *P. falciparum* and 9h in *P. knowlesi*. During this period, the nuclei in a *P. knowlesi* schizont completed a maximum of 4 and a mean of 2–3 replicative rounds, whereas *P. falciparum* completed a maximum of almost 5 and a mean of 3–4 replicative rounds. *P. falciparum* was therefore proportionally less efficient, averaging an extra 40–45 minutes per nuclear multiplication. This was primarily due to longer gap phases between replicative rounds, since the genomes are very similar in size (~23 and 24Mb), single-molecule dynamics were not strikingly different in the two species, and active DNA replication periods were not consistently faster in *P. knowlesi*–rather, they were slower in the first replicative round, and then faster in later schizogony.

There are several explanations for the longer pauses between replicative rounds in *P. falciparum*. This genome may be particularly challenging to replicate, with its high A/T-richness and prevalence of homopolymer tracts leading to errors that require resolution before the chromosomes can be separated. If this is true, a 'G2 checkpoint' (well-characterised in mammalian cells but uncharacterised in *Plasmodium*) must exist. The gap after karyokinesis but before the next replicative round was also longer in *P. falciparum* and this could be imposed, for example, by a certain amount of polymerases or nucleotides being required before a

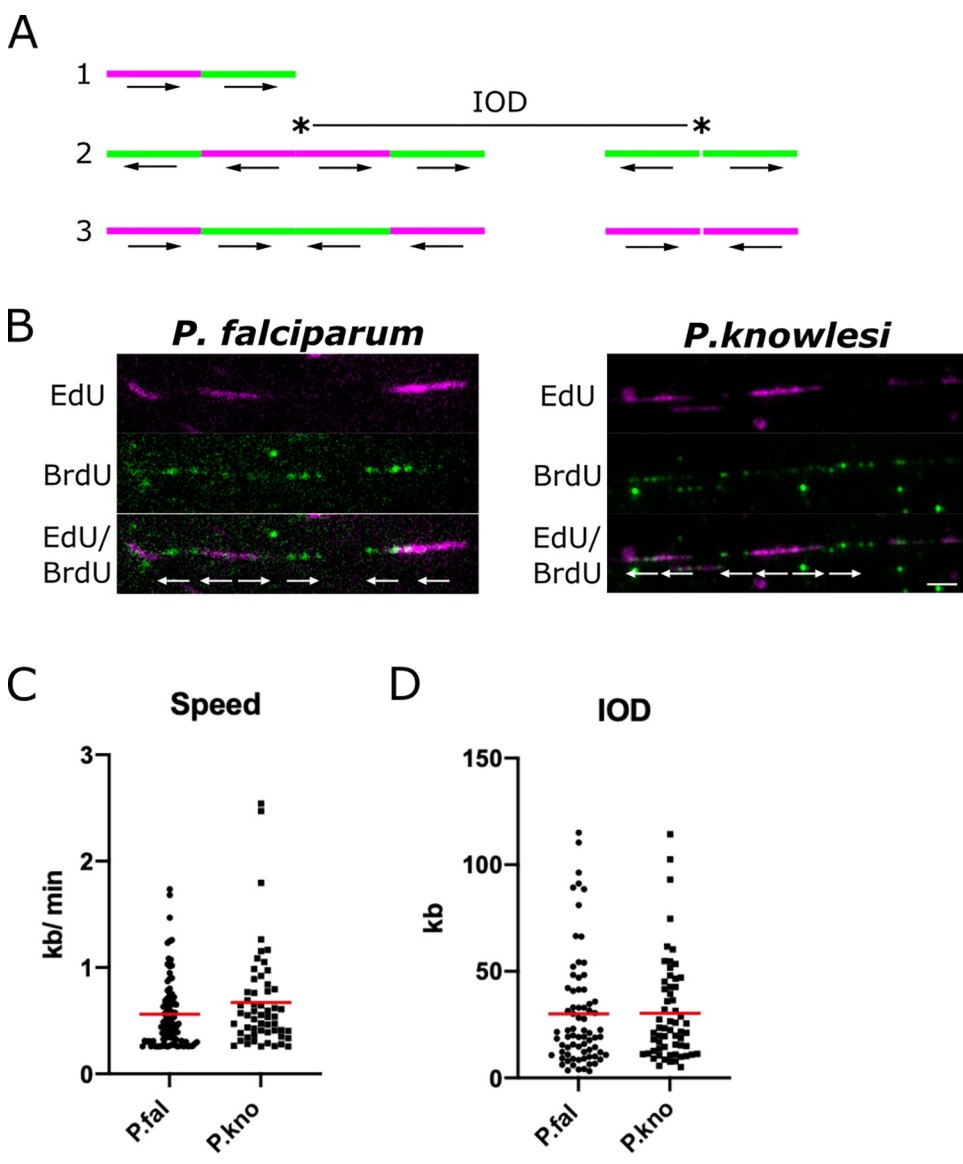

**Fig 7. Single-molecule DNA replication dynamics in *P. falciparum* and *P. knowlesi*.** A: Schematic showing possible patterns of labelling on DNA fibres. Pattern 1 is an ongoing replication fork, moving throughout the two consecutive pulse-labels. Pattern 2 shows origins that fired during the first and second pulse labels, with inter-origin distance (IOD) measured as centre to centre distance between two adjacent origins. Pattern 3 shows two replication forks terminating during the first or second pulses. The presumed positions of the replication origins are indicated with asterisks in the middle of bidirectional replication forks. Arrows represent the direction of the replication forks' progression. B: Representative DNA fibre spreads from *P. falciparum* and *P. knowlesi*. A 5kb scale bars is indicated. C: Dot plot showing distribution of replication fork speeds in *P. falciparum* and *P. knowlesi*, calculated on the basis of a 20-min pulse label (10 mins EdU, 10mins BrdU) and a stretching factor of 2.59kb/µm. Red bars represent the mean value. D: Dot plot showing distribution of inter-origin distances (IODs) in *P. falciparum* and *P. knowlesi*, calculated on the basis of a stretching factor of 2.59kb/µm. Red bars represent the mean value.

nucleus can trigger DNA replication, and this threshold being slower to achieve in *P. falciparum* than in *P. knowlesi*. Interestingly, a 'limiting factor' hypothesis has also been proposed from orthogonal recent work that is discussed in more detail below [11].

During S-phase the number of genomes in a schizont dramatically increases, but surprisingly there was no set-point in the proportion of genomes that could replicate DNA at once.

Schizonts were observed with anything from 20% to 100% of their nuclear masses replicating in a 30-minute window, with the average proportion being higher in *P. falciparum* than in *P. knowlesi*. Thus, the supply of DNA replication factors must keep pace with the expanding number of genomes–indeed, replicative rounds actually grew faster not slower. The growing anabolic challenge for an unknown 'limiting factor' may, however, be one reason why the gaps between replicative rounds increased as schizogony progressed. It may also account for why some nuclei appeared to arrest for much longer than average (either within DNA replication or at anaphase). A second explanation for arrested nuclei would be unresolvable DNA damage, triggering a checkpoint arrest, although there is little evidence as yet for conventional replication checkpoints in this system. (Halogenated nucleotides can also cause light-sensitive DNA damage, but cultures were kept in the dark at all possible times to minimise this risk.)

Both species were strikingly variable in the number of daughter cells produced per schizont. Such variation has been observed repeatedly in *P. falciparum*, with mean merozoite numbers of 15–22 reported in different strains [12, 17, 24, 25]. This cannot be entirely due to asynchrony in cultured cells: for example, the majority of *P. knowlesi* cells entered S-phase within a single hour, yet nuclear numbers were extremely variable at 31hpi, when just 2h later the great majority of these schizonts would burst. (Asynchronous S-phase entry may play a somewhat greater role in *P. falciparum*, where the pre-S-phase growth period was 50% longer and S-phase entry was less tight.) One key explanation for variable schizont size could be heterogenous long-term stalling of some nuclei, as shown in Fig 6: if, at the 2n stage, one nucleus arrests, then the schizont might complete only half as many nuclear multiplications and this would logically lead to highly variable schizonts. Another explanation is inherent variability in host erythrocytes, with young or old cells providing more or less hospitable environments for parasite multiplication. (It was also notable that human erythrocytes from different donors, which can differ in age and many other factors, had a strong impact on cell-cycle length, particularly in *P. knowlesi*, so cell-cycle length can certainly be influenced by host as well as parasite factors.) Regarding parasite factors, different strains of *P. falciparum* are known to have characteristically variable cycle times, ranging from ~48h in 3D7 [26] to 42-44h in FCR3 [2]. In this work, only the 3D7 strain was studied.

Our data are largely consistent with data from Ganter and colleagues, who recently characterised S-phase dynamics in *P. falciparum* in a complementary way, using live-cell microscopy with the DNA replication factor PCNA[11]. Their method is independent of DNA labelling, and therefore of any associated artefacts such as DNA photo-damage, yet the S-phase parameters measured were strikingly similar. They reported that in early schizonts active DNA replication (marked by the presence of PCNA) takes ~40–50 mins per nucleus, with a broad range in different nuclei and a strong trend towards longer periods in the first replicative round. The gaps between rounds, both before and after karyokinesis, were much longer that DNA replication itself, and were longer at the 1-to-2n stage (~75 mins before and ~50 mins after karyokinesis) than at the 2-to-4n stage. Cells beyond this stage were not analysed. These data are broadly consistent with ours: that early schizonts tend to take 40–75 minutes per active replicative round and ~3h in total between successive nuclear multiplications. They do not address our additional finding that gaps become longer in mature schizonts, but these authors did generate a mathematical model for overall schizogony, which demanded a slowing of ~17% in each replicative round after the second. To comply with this, if DNA replication itself successively speeds up then gap phases must become longer: consistent with our data, and also with a requirement to build up a putative limiting factor. Ganter and colleagues also proposed this from their independent dataset, strongly supported by their observation that when several nuclei replicated their DNA synchronously, they took longer to complete.

It was striking that centrin foci, representing the centriolar plaques that act as microtubule organising centres for each genome division, were only detected with ~50% of all nuclear masses in schizonts of either species. These data may have been affected by use of a sub-optimal commercial antibody [18], but nevertheless it is likely that centriolar plaques are not static, but are elaborated and disassembled in at least some components with each replicative round. They clearly disappeared in late schizonts after the final round of karyokinesis, consistent with Simon *et al.* [18] and expected if the structures get disassembled when no longer required. Furthermore, only a small minority of nuclear masses showed two well-separated plaques, suggesting that their duplication and separation is a late event in the process of DNA replication and karyokinesis. A caveat here is that centriolar plaques were detected only by centrin, which may be one dynamic component of an underlying, static structure. The detailed organisation of the *Plasmodium* centriolar plaque remains under active investigation [18].

Although *Plasmodium* schizogony is a very different process from binary fission in mammalian cells, the intra-nuclear organisation of genome replication appears to be similar. Mammalian nuclei show foci of active DNA replication called 'replication factories'. In early S-phase, when euchromatin is predominantly replicating, factories are small, numerous and dispersed throughout the nucleus; later they appear as larger clumps in the nuclear periphery, representing replicating heterochromatin [27]. *Plasmodium* nuclei are comparatively tiny but it was nevertheless possible to see brighter foci of newly replicated DNA in the nucleoplasm, resembling replication factories. (These were particularly evident in nuclei stained with BrdU, which was detected less strongly than EdU, using an antibody rather than click-chemistry, thus emphasising the strongest foci.) A small proportion of nuclei showed very clear perinuclear foci, possibly because they were captured while replicating discrete perinuclear clusters of heterochromatin, and some showed DNA replication in only one section of the nucleoplasm–these were much commoner in *P. falciparum* than *P. knowlesi*. Chromosomes may be organised differently, or heterochromatin may be more strongly clustered, in *P. falciparum*, reflecting the well-characterised clustered heterochromatin that encodes *var* virulence genes [28]. Finally, during the presumptive final round of DNA replication, there was a clear pattern of homogenous but faint staining in all nuclei: this was particularly dominant in *P. falciparum*. It suggests that the final round is indeed distinct: DNA replication in all nuclei may slow down, either to ensure synchrony in the final karyokinesis, or because a certain factor becomes limiting. Such low-level DNA synthesis could, alternatively, represent post-replication DNA repair, but this seems unlikely because it was usually seen throughout the nucleoplasm of all nuclei, and quite specifically in late schizonts.

At the single-molecule level, there was little difference in the dynamics of polymerase movement (i.e. replication fork velocity) through the genomes of *P. falciparum* and *P. knowlesi*, although there was a trend towards faster movement in *P. knowlesi*. This could be due to genome composition, or to a higher average level of DNA replication factors (proteins or nucleotides) in *P. knowlesi* schizonts. DNA replication dynamics do frequently differ between different species and even between different human cell lines [29]. Here, however, it seems that the main differences between *P. falciparum* and *P. knowlesi* occur at a higher level–i.e. the timing of multiplication for whole nuclei and their associated gap phases–rather than at the level of individual DNA replication forks. It is interesting to speculate the wide variation in replication fork rates may originate partly from pooling cells in which both many and few nuclei are replicating simultaneously, and accordingly where a 'limiting factor' is more or less limiting per replication fork.

Overall, this study provides novel detail about the dynamics of schizogony in two different human malaria parasites, and sets the stage for future work to examine how schizogony changes with changing conditions in the human host. A recent publication reported that

nutrient limitation in a murine host can markedly reduce the number of progeny per schizont, and suggested that the same may occur in human malaria patients [30]. This may be only one of many host conditions that could influence the growth of malaria parasites, with potentially important impacts on clinical outcome.

## Methods

### Parasite culture and transfection for ectopic expression of thymidine kinase

*P. falciparum* parasites were maintained *in vitro* in human O+ erythrocytes at 4% haematocrit in RPMI 1640 medium supplemented with 25mM HEPES (Sigma-Aldrich), 0.25% sodium bicarbonate, 50 mg/L hypoxanthine standard procedures [31]. *P. knowlesi* parasites were maintained similarly, maintained at 2% haematocrit instead of 4% and supplemented with 22.2 mM glucose and 10% horse serum instead of human serum.

The *P. falciparum* 3D7 strain that expresses thymidine kinase has been previously described [10]. A similar thymidine-kinase-expressing *P. knowlesi* strain was created by transfecting the same plasmid into the A1-H.1 strain, essentially as previously described [32]. Late-stage *P. knowlesi* parasites were enriched using Histodenz and 10 μl of schizonts mixed in a transfection cuvette (Lonza) with 100 μl of P3 solution (Lonza) containing 30 μg of plasmid. Transfection was carried out using program FP158 (Amaxa Nucleofector, Lonza), followed by immediate transfer into 500 μl of complete culture media mixed with 190 μl uninfected erythrocytes. The transfection mix was incubated at 37°C while shaking at 800 rpm in a thermomixer for 30mins, before being transferred into a 6-well plate, gassed and incubated for one parasite life cycle. Selection was then applied with 100 nM pyrimethamine (Santa Cruz Biotechnology Inc) and daily media changes for 3 days, then routine maintenance until transgenic parasites appeared.

### Synchronisation for timecourse experiments

Mature schizont cultures at >6% parasitaemia were synchronised using 55% Nycodenz (Alere technologies AS) [33]. Cultures were centrifuged and media removed to leave 2ml of media and blood, which was layered gently on top of 5 mL of prewarmed 55% Nycodenz, then centrifuged at 1300g for 5 mins. The floating schizont layer was collected and added to a wash buffer containing incomplete RPMI, 4% haematocrit for the final culture volume, and 1.5 μM 'Compound 2' (4-[7-[(dimethylamino)methyl]-2-(4-fluorophenyl)imidazo[1,2-*a*]pyridine-3-yl]pyrimidin-2-amine) [34]), centrifuged (800g for 5 mins), resuspended in complete RPMI and 1.5 μM Compound 2 and incubated at 37°C for 2 h. Cultures were centrifuged (800xg for 5 mins) and supernatant removed. Cultures were washed in prewarmed incomplete media and resuspended in complete RPMI to allow reinvasion. Cultures were split into 5ml aliquots in 50ml falcon tubes and placed in an orbital shaker at 37°C for 1 h to increase reinvasion rate. After reinvasion, the cultures were pooled together, centrifuged and Nycodenz treated again, now retaining the bottom layer containing newly reinvaded ring stages. This layer was washed in incomplete media and resuspended in compete RPMI, marking timepoint 0 hours post invasion (hpi).

### Pulse labelling with modified nucleotides

Cultures for single-labelled timecourses were pulse-labelled with 10 μM ethyl-deoxyuridine (EdU) for 30 mins at 1 h intervals. Cultures for double-labelling experiments were first pulse-labelled with 10 μM EdU for 30 mins at specific timepoints, then washed twice in prewarmed

incomplete RPMI and resuspended in complete media that was incubated alongside the cultures throughout the experiments, to avoid any disturbance of cell cycle dynamics caused by switching into fresh media. At specific timepoints, cultures were labelled with a second nucleotide, 5-bromo-2'-deoxyuridine (BrdU), at 200 μM for 30 mins at 37˚C. Immediately after this label, blood smears were made, air dried, fixed in 2% paraformaldehyde for 5 mins, washed in PBS, washed in $dH_20$, air dried and stored at 4˚C.

## Immunofluorescence

All slides were incubated in 0.2% Triton X-100 for 15 mins and washed in PBS for 5 mins.

Single-nucleotide EdU-labelled slides were incubated in blocking solution (1% BSA in PBS) for 30 mins and then 1:100 anti-centrin antibody (clone 20H5, Millipore) in blocking solution for 1 h at room temperature. Slides were washed three times in blocking solution. EdU signal was detected with click chemistry. Slides were incubated in click reaction buffer (0.845mM Tris HCl pH 8.8, 1mM $CuSO_4$, 2.5 μM Alexa Fluorescent Azide 594, freshly dissolved 75mM ascorbic acid) for 1 h at room temperature. Slides were washed in blocking solution three times and then incubated with the secondary antibody for centrin detection, goat anti-mouse Alexa 488 (Molecular Probes) for 1 h at room temperature. Slides were washed twice in PBS, incubated with 2 μg/ml 4′,6-diamidino-2-phenylindole (DAPI) for 10 mins, washed in PBS, mounted using 20 μl Prolong Diamond Antifade (Molecular Probes) and set overnight at room temperature.

Double-labelled EdU/BrdU slides were incubated with 0.2% Triton-X for 15 mins, washed in PBS for 5 mins, incubated in 1M HCl for 1 h and washed in PBS for 5 mins. EdU labels were "clicked" as above. Slides were washed with PBS three times. Slides were then incubated with 20 mM non-fluorescent dye Azidomethylphenosulphide (Sigma) for 30 mins at room temperature to block remaining EdU residues. Slides were incubated with blocking solution for 30 mins. Primary immuno-detection of BrdU was with rat anti BrdU BU1/75 (ICR1) antibody (1:100 dilution, Abcam). Slides were washed in blocking solution three times and incubated with goat anti-rat Alexa 488 secondary antibody (1:500 dilution, Molecular probes) at room temperature for 1 h; washed, incubated with DAPI and mounted as above.

## Data analysis and statistics

Slides were imaged using a Nikon Microphot SA microscope equipped with a Qimaging Retiga R6 camera and a 100X oil objective (Leica, 1.30 na). 100 parasites were counted for each timepoint of the single-nucleotide-labelled timecourses, and 50 parasites were counted for double-labelled timecourses. For the 'stalling' experiment, 20 parasites were counted for each category of schizont maturity. Images were classified regarding presence and number of centrin foci and the presence and pattern of nucleotide labelling (BrdU/EdU) within the nucleus. Widefield microscopy was used throughout because very large numbers of parasite images were required, accepting that the size of individual nuclear masses approaches the limit of resolution for simple light microscopy. (See S2B Fig for a comparison of single-projection versus confocal microscopy when used for counting nuclear masses. Confocal imaging was carried out using a Zeiss LSM 700 microscope using Zen10 software). Data were plotted using Graphpad Prism and the statistical significance of differences between groups of data was calculated via Mann Whitney tests or analysis of variance.

## DNA fibre spreading

DNA fibre spreading was performed as previously described [22]. Cultures were labelled with 10μM EdU for 10 mins, then 100 μM BrdU for 10 mins. 2μl of saponin-released parasites was

pipetted near the top of a glass slide and allowed to dry for ~5 mins, until sticky but not dry. 7 µl of spreading buffer (20 mM TrisHCl pH 7.4, 50 mM EDTA pH 8, 0.5% SDS) was added and stirred gently with a pipette tip to release the DNA. The slide was incubated for 2 mins and then tilted (15˚) to let drop run slowly down the slide, producing a constant stretching factor of 2.59kb/µM [23]. Slides were air dried, fixed in MeOH: acetic acid (3:1), air dried and stored at 4˚C.

## Detection of EdU and BrdU in DNA fibre spreads

Fibre spreads were denatured with 1M HCl for 75 mins, washed three times in PBS and blocked with blocking solution (1% BSA and 0.1% Tween 20 in PBS). Immuno-detection of double labelled fibre spreads was done with EdU click chemistry and then antibodies diluted in blocking solution, each incubated with a coverslip on top in a humid chamber at room temperature for 1h. Slides were incubated with click reaction buffer as above, followed with three washes in blocking buffer. Slides were then incubated in blocking solution for 30 mins. Primary immuno-detection for BrdU was done with rat anti BrdU BU1/75 (ICR1) antibody (1:100 dilution, Abcam), together with mouse anti ssDNA (clone16-19) antibody (Millipore, 1:300 dilution). The secondary antibodies (Molecular Probes) were goat anti-rat coupled to Alexa 488 (1:500 dilution) and goat anti-mouse coupled to Alexa 405 (1:500 dilution). Slides were washed three times in PBS and mounted using 20µl Prolong Diamond Antifade (Molecular Probes), set overnight at room temperature. Single-labelled fibre spreads were detected above but without click chemistry.

## Image Acquisition and Processing of DNA fibre data

Image acquisition was via a Nikon Microphot SA microscope equipped with a Qimaging Retiga R6 camera. Images were acquired with a 100X oil objective (Leica, 1.30 na) where 1 µm = 28.65 pixels, which corresponds to 69.8 bp per pixel (DNA stretching factor 2kb/ µm for DNA combing) and 90.40bp per pixel (DNA stretching factor 2.59kb/µm for DNA spreading). Observation of long DNA fibres required the capture and assembly of adjacent fields. Replication tracts and fibre lengths were measured manually using ImageJ software. Statistical analysis and graphs of BrdU tract length and replication velocities were performed using GraphPad Prism.

## Availability of raw data

All data are available in the Dryad repository at doi:10.5061/dryad.ghx3ffbr8 [35].

## Supporting information

**S1 Fig. DNA replication rates are similar in the newly-generated *P. knowlesi* TK-expressing line and in the parent line.** Synchronised parasites of both lines were exposed to either or both modified nucleosides at the levels used in subsequent experiments (10 µM EdU, 200 µM BrdU) for a 6h period covering the majority of S-phase. DNA content was then measured via SYBR-green 1 DNA dye, as previously published [36], in triplicate, and expressed as fold-change in DNA content from the start of the experiment. No significant difference in growth rates was observed in either line, exposed or not exposed to modified nucleosides.
(TIF)

**S2 Fig.** Examples of A) a typical slide, showing 3 parasites classified for their number of nuclear masses, number of centrin foci and presence/absence of EdU staining. B) confocal microscopy versus the wide-field method used throughout this work, showing that confocal

yields very similar detection of the number of features per cell.
(TIF)

**S3 Fig. Biological replicates of timecourses shown in Fig 2.** In the early parts of the time-courses, samples were taken every 3h for *P. falciparum* and every 2h for *P. knowlesi*, rather than every hour, as in Fig 2. In the final parts of both timecourses, samples were again taken every hour. Final timepoints in both timecourses show reinvasion, i.e. high proportions of 1n cells. A, B: Scatter graphs of nuclear numbers in *P. falciparum* (A) or *P. knowlesi* (B), n = 30 cells, medians are shown in red. C, D: Percentage of 30 cells showing some EdU labelling at each timepoint in *P. falciparum* (C) and *P. knowlesi* (D).
(TIF)

**S4 Fig. Data from Fig 3D (i.e. intensity of EdU staining seen at each timepoint in *P. falciparum* and *P. knowlesi*) shown as individual graphs for each category of staining defined in Fig 3B.**
(TIF)

**S5 Fig.** A: Schematic showing the process of karyokinesis that has previously been proposed in *Plasmodium*, highlighting the role of the centriolar plaque (basis outlined in Gerald *et al.* [13]). B: Examples of the distinct patterns of centrin staining seen on the highlighted nuclear masses: no foci, a single focus, 2 adjacent foci, 2 opposite foci. *P. falciparum* are shown, representative of pattern in both species. Scale bar all panels 1μm. C: Percentage of nuclear masses with 0, 1 or 2 centrin foci throughout schizogony, n = 100. D: Percentage of nuclear masses with 0, 1 or 2 centrin foci that also showed or did not show active DNA replication (EdU staining) within the previous 30mins. E: Percentage of nuclear masses with 0, 1 or 2 centrin foci and patterns of intranuclear DNA replication (full, partial, or discrete foci) throughout schizogony. F: Data as in Data as in A, replotted by number of by number of nuclear masses per cell rather than hpi.
(PNG)

**S6 Fig. The minimum pulse-labelling period that can be detected via EdU-labelled DNA was tested.** A: In *P. falciparum*, 3 minutes was detectable (faintly) and 5 minutes gave reasonably bright signal. In *P. knowlesi*, labelling was clearly detectable within 1 minute. B: Examples of the appearance of stained nuclei when cells are simultaneously labelled with EdU and BrdU. Scale bar all panels 2μm.
(TIF)

**S7 Fig. Biological replicate of the experiment shown in Fig 4, measuring the length of the first replicative round in *P. falciparum* and *P. knowlesi* cells.** Graphs show the percentage of nuclei in 1n cells that labelled with EdU and BrdU (n = 20). Scatter plots display the same data broken down per-cell, with means shown.
(TIF)

**S8 Fig.** A: Schematic of the timecourses shown in this figure. B: Representative examples of cells across the double-labelled timecourse with 2h intervals for *P. falciparum*. Scale bar all panels 2μm. B: Percentages of nuclear masses labelled with EdU alone, BrdU alone, or both labels throughout the timecourses shown in (B).
(TIF)

**S9 Fig. Data as in main Fig 6, showing the percentages of *P. knowlesi* nuclei labelled with EdU and BrdU, EdU alone, BrdU alone, or neither label (n = 20 cells), after both 2h and 5h –demonstrating a higher percentage of arrested nuclei after 2h (~22% of S-phase) than 5h**

(>**50% of S-phase), but an overall similar picture.** Data are also stratified into cells with 2–3, 4–6, or more than 6 nuclear masses, showing that arrested nuclei are still detected in very young 2-3n schizonts.
(TIF)

## Acknowledgments

We are grateful to the lab of Prof. Julian Rayner for making the initial thymidine kinase transfectant in *P. knowlesi*, to Dr Holly Craven for help with revisions, and to Dr Francis Totanes and Dr Andrew Blagborough for critical reading of the manuscript.

## Author Contributions

**Conceptualization:** Catherine J. Merrick.

**Data curation:** Jennifer McDonald.

**Formal analysis:** Jennifer McDonald.

**Funding acquisition:** Catherine J. Merrick.

**Investigation:** Jennifer McDonald.

**Methodology:** Jennifer McDonald.

**Project administration:** Catherine J. Merrick.

**Supervision:** Catherine J. Merrick.

**Visualization:** Jennifer McDonald.

**Writing – original draft:** Jennifer McDonald, Catherine J. Merrick.

**Writing – review & editing:** Catherine J. Merrick.

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
