## [Decision Letter · Decision Letter 0]

14 Feb 2022

Dear Catherine,

Thank you very much for submitting your manuscript "DNA replication dynamics during erythrocytic schizogony in the malaria parasites Plasmodium falciparum and Plasmodium knowlesi" for consideration at PLOS Pathogens. As with all papers reviewed by the journal, your manuscript was reviewed by members of the editorial board and by several independent reviewers. In light of the reviews (below this email) and following extensive consideration, we would like to give you the opportunity to submit a significantly-revised version that addresses or takes into account the reviewers' comments.

As you will see, the reviewers reached widely divergent views on the manuscript. Reviewers 1 and 2, however, raised significant concerns that need to be addressed before the work can be considered further. In particular, questions were raised over the impact of BrdU and EdU labelling on cell cycle progression in the parasite lines, the significance of the apparent loss of centrin signal from nuclei, the number of experimental biological replicates used to draw several of the key conclusions, and whether the light microscopic system used provides sufficient resolution to reach some of the conclusions drawn. We share these concerns. All the reviewers additionally raised a number of additional points that should be addressed.

We cannot make any decision about publication until we have seen the revised manuscript and your response to the reviewers' comments. Your revised manuscript is also likely to be sent to reviewers for further evaluation.

Sincerely,

Michael J Blackman

Associate Editor

PLOS Pathogens

Xin-zhuan Su

Section Editor

PLOS Pathogens

Kasturi Haldar

Editor-in-Chief

PLOS Pathogens

orcid.org/0000-0001-5065-158X

Michael Malim

Editor-in-Chief

PLOS Pathogens

orcid.org/0000-0002-7699-2064

Reviewer's Responses to Questions

**Part I - Summary**

Reviewer #1: The authors provide an interesting description of replication dynamics throughout erythrocytic schizogony of two Plasmodium species. The amount of data generated is remarkable and the used technology is interesting. However, the study remains descriptive, and the draw conclusion do not extend significantly enough upon the current understanding of replication. The gain of sustainable insight and the ability to generate new hypotheses is a bit too limited to warrant a publication in a journal like PLoS Pathogens. Further, there are some important technical concerns that would have needed to be more thoroughly addressed by the authors to consolidate their data. Lastly the manuscript itself suffers from a difficulty to understand the individual experimental parameters and outcomes. Given some improvements I, however, know that this study can find its place in another journal. In the following I want to outline my criticism in more detail to help the authors to improve the manuscript.

Reviewer #2: In a patient infected with Malaria, Plasmodium parasites replicate in red blood by an atypical cell division mode called schizogony. Multiple rounds of asynchronous DNA replication and nuclear fission generate a multinucleated cell and, specialized cytokinesis named segmentation produces 10 to 30 daughter parasites depending on the Plasmodium species. As McDonald et al. highlighted in their study, the rate and efficiency of parasites replication are fundamentally critical to malaria pathogenesis. In the case of malaria infection by P. falciparum, peripheral parasitemia will determine the severity of the disease. To better understand the molecular mechanisms regulating the number of daughter parasites, the study's authors took a very original and ambitious approach by studying schizogony in two different human Plasmodium parasites, falciparum, and knowlesi, that differ in their cell cycle length and the number of progeny. The authors successfully generated parasite lines that allow de novo DNA replication to be followed by incorporating pulse labels of modified nucleotides. The authors performed different time courses using single EdU pulse labels or sequential pulses of two nucleotides (EdU and BrdU). All experiments readouts were done by light microscopy, except for the determination of the replication fork speed on spreading DNA fibers from cell lysates on glass slides. Unfortunately, I have significant concerns regarding the experimental robustness of the study, where it appears that for all data presented in the study, there is only a single biological replicate. In addition, the authors' microscopy technique does not provide the required resolution to resolve two nuclei, especially in the z-axis when they analyze multinucleated with more than four nuclei. Therefore the authors can observe and measure DNA replication per cell but not per nucleus.

Furthermore, they wrongly interpreted the published data on the MTOC dynamics in Simon et al.'s study, leading them to inappropriately conclude that 50 % of nuclei lack centrin staining, likely representing false negatives. In conclusion, I have raised significant comments that need to be addressed, and numerous statements must be revised based on the limitation of the experimental settings. In addition, I made some minor comments to help the authors identify sections of the study that remain unclear to me.

Reviewer #3: This work offers a detailed and comparative analysis of DNA replication dynamics of two distinct malaria parasite species. P. knowlesi and P. falciparum. It is of particular importance as the first paper for which the replication dynamics of Pk are described, and uses a series of neat approaches to compare the specific replication parameters. This comparison is particularly important as the two parasites have significant differences in cycle length (Pf 2x as long) and AT-content, as well as ultimately producing differing numbers of progeny merozoites. The studies revealed surprising similarities in the single molecule speed of replication as well as inter origin distance, but some clear differences in whole cell replication dynamics – with a surprisingly synchronous onset of S phase and shorter gap phases.

Although descriptive the data presented are of fundamental importance to understanding these processes, and thus I would say they are both highly important for work in the field, and would provide significant interest to the pathogen research community. The paper is very well written and the conclusions are well supported by the data. In the current version, it is slightly unclear as to what biological repeats have been carried out for each experiment, and this is something that will need to be clarified. The other issues raised are relatively minor, although efforts to examine perturbations of these dynamics would add another level of interest.

Rob Moon

**Part II – Major Issues: Key Experiments Required for Acceptance**

Reviewer #1: The authors don´t comment sufficiently on the very low mean number of nuclei at the end of schizogony (P.f. 11 and P.k. 6) shown in Fig. 2. This stands in contrast to what has been measured in several studies by e.g. Garg et al. , Reilly et al., Simon et al. and would warrant a more detailed analysis. The observation that number of nuclei is highly variable is not novel as suggested in the abstract.

The authors claim that “The length of S-phase is proportional to the overall length of erythrocytic schizogony”. I think their data shown in Fig. 2 is informative overall and underpins that schizogony takes about 30% of the cycle. This, however, shouldn’t be mistaken for a correlation between total cell cycle length and schizogony at the single cell level and clarified accordingly.

The authors claim that centrin signal temporarily disappears from a majority of nuclei and suggest this represents a centriolar plaque disassembly. To my knowledge temporary centrosome disassembly (down to centrin) during a division phase is not described in any eukaryote. For Plasmodium specifically multiple studies investigating centrin don’t report any similar findings although their imaging data quality is good (Arnot et al. 2011, Roques et al. 2019, …). Particularly the Simon et al. study referenced by the authors, as far as I understand does only refer to 50% of mononucleated nuclei lacking a centrin signal while in multinucleated stages all nuclei carry one or two centrin foci. Finally, the signal-to-noise ratio shown in Fig. 4B casts significant doubt on the validity of this rather bold claim.

Most importantly, the authors fail to demonstrate that cell cycle progression was “not overtly affected” (line 117) by their labeling. This is particularly critical in the light of what the last author has shown in Merrick et al. 2015 where TK expressing parasites displayed a high sensitivity to BrdU (and possibly EdU?) and a severe growth phenotype at significantly lower concentrations than used in this study (IC50 of 219nM measured for a 1 hour pulse – against - 200µM pulse for 30min used here). Unfortunately, this puts into question the interpretation of a big proportion of the data and the authors have done very little to alleviate this concern. BrdU toxicity could for example be the cause for the replication gaps observed for a subpopulation of nuclei in Fig. 6. This is probably even more critical for the observations made in Fig. 7 that nuclei seem to pause replication for extended periods of time.

Reviewer #2: General

My main concern is that it appears that for all data presented in the study, there is only a single biological replicate, in particular for assays run with P. falciparum. Whether there are multiple biological replicates or only one should be specified for every experiment in the figure legend. Suppose there is only a single biological replicate. In that case, the authors should provide extensive justification for why this is the case, and this should be reflected by significantly dampening the corresponding conclusions. The authors note the difference in observed lifecycle length for P. knowlesi based on different batches of donor blood and in different P. falciparum 3D7 parasite line ranging from ~39 hours to the ~48 hours reported in this study (Wockner et al., 2020 (PMID: 31679015); Duffy and Avery, 2017 (PMID: 28738214). Considering the potential influence of lifecycle length on the estimation of the length of S-phase.

In line 184, the authors rigorously and rightfully used the term DAPI-stained area instead of nucleus. The resolution is calculated by the following formula Resolution=1.22 (λ/(NAobj + NAcond)), if we assume that the authors used an objective with a high numeric aperture or NA (1.45) and a condenser of 0.55, the resolution for the DAPI channel (405 nm) is as best 247.5 nm for the lateral resolution (resolution in the focal plane). Therefore in this study, the resolution is four times bigger than an individual nucleus (200nm at best versus 50 nm); consequently, the authors cannot identify single nuclei in a multinucleated cell and must revise their results statements.

The data from the referenced Simon et al. study are inappropriately referenced to suggest that detection of centrin in ~50% of nuclei is consistent with previous literature. This was only the case in mononucleated parasites with a hemispindle, with the following statement made in that article: “Consistent with the late appearance of PfCentrin1-GFP in our live-cell imaging data, only 24 out of 52 analyzed hemispindles in mononucleated cells were associated with an endogenous centrin signal, whereas in later stages, after the first division, every nucleus was accompanied by one or two centrin foci”. Presumably, from 33-47hpi in P. falciparum and 23-33hpi in P. knowlesi, you have relatively few mononucleated parasites. Therefore the ~50% of nuclei lacking centrin staining likely represents a vast number of false negatives. Finally, the anti-centrin antibody labels only one protein present at the cytosolic face of the MTOC, which is a large multiprotein structure embedded in the nuclear envelope for which we know very few components. The authors should address this and consider this when making conclusions about centrin and MTOC.

Results

Line 115: “We confirmed that this permitted the labelling of DNA replication with modified nucleotides (bromo/chloro/iodo/ethyl-deoxyuridine (BrdU/CldU/IdU/EdU)), and that parasite behaviour in terms of fitness and cell-cycle timing was not overtly disturbed". Data is not shown or unclear where the authors presented it in the manuscript.

Line 133-135: the number of nuclei detected at the latest timepoints for schizonts are drastically lower than previously published numbers of merozoites per schizont, at least for P. falciparum where this has been well characterized (Garg et al., 2015 (PMID: 26702305); (Rudlaff et al., (PMID: 32511279)). The authors should elaborate on why this is the case, be it an issue of resolution, parasite maturity, or something else, and clearly state this when making conclusions based on the number of nuclei per cell.

Line 139: “…the onset of S-phase for the majority of cells was quite synchronous (Figure 2C, D) particularly in P. knowlesi, where ~60% of cells commenced replication within a single hour at 22-23hpi.” This doesn’t seem to be supported by the data in Figure 2C & D. According to the methods, these cultures are synchronized to within 1 hour of each other, yet there is a 4 hour difference between when the first parasites begin to replicate (22hpi) and when the greatest number of parasites are replicating (26hpi). This could be interpreted as up to 4 hours of replication asynchrony within a population that should be within 1 hour of age.

Line 145: “Numbers of nuclei stopped increasing at 46-47hpi, making 15 hours the total S-phase period across the population.” It seems in Figure 2C that ~80% of cells are EdU positive at 47 hours, indicating that at 47 hours, 80% of cells are still replicating and could very well increase in DNA content resulting in the formation of new nuclei past the 47 h.p.i. Considering that 80% of parasites were still replicating at this time point, it is difficult to see how it can be used to conclude the length of the total S-phase. The measure of the total content of DNA would be a more appropriate measure than many nuclei, which again cannot be resolved with the microscopy technique presented in the study without treating the parasite with E64, which would allow to increase in the physical distance between nuclei and allow their counting by light microscopy. Therefore the data presented in Figure 2, panels A, B, C, F, G, and H must be revised and interpreted accordingly.

Line 153-154: “P. falciparum showed substantial reinvasion by 48hpi (Supp Figure 2), and so has a similar window of ~2h between the end of S-phase and reinvasion.” Supp Figure 2 doesn’t appear to have any data regarding EdU labelling, and so can’t be used to support the claim that S-phase has ended.

Line 186-189 “…particularly notable in P. falciparum, was a subset of nuclei with homogenous but fainter staining, suggesting that DNA had replicated much less rapidly than in the common pattern: this appeared mostly in late-stage cells completing their presumptive final round of replication (Figure 3B, see P. falciparum at 44-45hpi, P. knowlesi at 30hpi, and quantification in Figure 3C, D)” There are many reasons why fluorescence intensity may differ between different cells, and it is unclear why DNA replicating more rapidly would be the reason for this difference. Also, your study and others suggest that DNA replication occurs faster at the later stages of schizogony, which contrasts with the idea that DNA had replicated less rapidly in these cells.

Line 210-212: “In both species there was some evidence that nuclei with centrin foci anti-correlated with actively replicating, centrin-negative nuclei see for example 42hpi in Figure 4D). These observations point to a temporal separation between the start of DNA replication and the start of centriolar plaque elaboration and duplication”. Considering the previously mentioned likely high false negative rate, and a lack of statistical tests showing this correlation, the data presented in Figure 4 is not sufficiently robust to support these claims. Additionally, it is unclear how you can infer temporal separation between DNA replication and MTOC without live-cell microscopy.

Line 249: “Therefore, replicative rounds became progressively faster as ploidy increased” In my understanding figure 5 supports the conclusion that DNA replication is asynchronous in both Plasmodium species and that is all. And because the experimental design does not allow live imaging therefore authors cannot assess the rate of round of DNA replication like it has been done in the preprint of Klaus et al prublished in 2021.

Discussion

Line 367: " The growing anabolic challenge may, however, be one reason why the gaps between replicative rounds increased as schizogony progressed. It may also account for why some nuclei appeared to arrest for much longer than average". The authors assess DNA replication in this study and not DNA segregation and karyokinesis. The authors call the arrested nucleus what might be an anaphasic nucleus when the two replicated interpolar microtubules connect DNA masses within the same nuclear envelope upon nuclear fission. It is unknown how long this step takes during plasmodium mitosis and whether or non this process is shorter in time as schizogony progresses.

Line 405: "centriolar plaques are disassembled with each replicative round". This relates to the previous comment regarding centrin not being an appropriate marker for the entire MTOC and the high false-negative rate. Supplementary Video 2 of the Simon et al. study shows first the first ~2-3 rounds of mitosis, where intranuclear microtubules are always associated with at least one centrin focus. Additionally, in another study where MTOC of P. falciparum was observed (using NHS ester instead of centrin), an MTOC was observed for every nucleus for every image included in the study (Liffner & Absalon, 2021 (PMID: 34835432), see supplementary videos).

Figures

While the median is often reported, there are no error bars presented on any of the data in the study. Therefore, the authors should include and define error bars for all data where multiple data points are reported.

Many of the microscopy images presented in this study lack scale bars or only one image in an entire panel of figures will have a scale bar. Therefore, either all images should include defined scale bars. If the imaged area is identical for all images in a particular panel, this should be stated in the figure legend that the same scale bar can be applied to all images.

For Figures 3a and 4b, it is currently unclear which organism these images represent. Therefore, the representative images should include examples of EdU and centrin staining classifications for both P. falciparum and P. knowlesi.

Methods

Based on the synchronization protocol listed in the methods, the authors seem to purify viable ring-stage parasites using Nycodenz. However, the authors do not reference a previous study establishing this protocol. If this protocol was developed in this study, the authors should provide validation data that it purifies ring-stage parasites or provide a reference if developed elsewhere.

The numerical aperture for the objective lens used in this study should be stated.

Reviewer #3: Major Points

The experimental design is laid out in Fig 1, and numbers of cells examined are mentioned in several figures but it is not clear what biological repeats have been carried out. Are cells examined all from the same cultures on same day? Or imaged from different repeat experiments. I would expect some parameters measured to vary depending on how “happy” a given culture was, and so data from independent cultures is vital. Could the authors clarify how biological repeats were undertaken, clearly stating this in each figure.

**Part III – Minor Issues: Editorial and Data Presentation Modifications**

Reviewer #1: Generally, I notice a “disconnect” between the findings claimed in the abstract, the sub-titles of the results and figures, and the conclusion made at the end of each sub-chapter of the results. This leads to difficulties in clearly grasping the main findings of the study.

Associating the schematics of the experimental setup directly with the corresponding figure would be very helpful to the reader to easier understand the complex data.

The category “fully stained, less bright” used in Fig. 3 bears some risk of not being clearly separatable from the background. Possibly a more objective criterion could be applied here. Maybe it is also helpful to stratify the integrated densities shown in Fig. 3D by the categories shown in Fig. 3C.

For Fig. 5 it would be good to show a simultaneous labeling with BrdU and EdU as a positive control.

The interpretation of the data collected in Fig. 6 could benefit from a more detailed explanation or visualization. Further, the authors could improve on their argumentation how their pulse labeling protocols distinguishes between replication length, replication gaps, and continuously, or not at all replicating nuclei in a multinucleated population.

The authors use the term “syncytial mode of replication” in the abstract. The word syncytium is, however, clearly defined as a multinucleated cell resulting from cell-cell fusion not from multiple nuclear divisions without cytokinesis (Daubenmire, 1936, Science and Wikipedia). Although I have noticed this term see casually applied for Plasmodium spp. I want to strongly advice against this usage.

Reviewer #2: Introduction

Line 47: “The malaria parasite Plasmodium”. This should be revised to say the genus of malaria parasites.

Line 95: the following is posed as an unanswered question “Do all successive rounds of nuclear replication take the same time?” This question has largely been addressed by the Simon et al., and Klaus et al., studies cited elsewhere in this paper.

I personally appreciate when authors conclude the introduction section by summarizing the major findings of the study, which is missing here.

The following statements should be qualified with references:

Line: 55 “Parasitaemia (Parasitemia) in human malaria can vary greatly, partly because the human-infective Plasmodium species (P. falciparum, P. vivax, P. malariae, P. knowlesi, and P. ovale wallikeri & curtisi) differ in their preferences for human erythrocytes.”

Line 58: “…P. vivax is restricted to invading scares reticulocytes, P. malariae primarily invades older erythrocytes, and P. falciparum invades cells of all…”.

Line 61: “There are also other inter-species differences in the process of schizogony. Its duration varies from ~72 hours in P. malariae to ~28 hours in P. knowlesi and the number of merozoites produced per schizont varies from as many as ~30 in P. falciparum to ~15 in P. knowlesi.”

Line 68: “…P. vivax and P. knowlesi have A/T contents of only 60 and 61%.”

Line 71: “These differ in parameters such as genome content, cell cycle period and merozoite number.”

Line 81: “…the first round of DNA replication begins more than halfway through the cell cycle (which is theoretically 48h, actually ~42-50h in different stains and culture systems).”

Line 85: “Duplication of the centriolar plaque (the Plasmodium centriole-equivalent) has been proposed to initiate each replication.”

Line 98: “At the single-molecule level, our recent work defined the average speed of replication forks and spacing of replication origins in P. falciparum schizogony at 1.2 kb/min and 65kb between origins – broadly similar to parameters in other eukaryotic cells. These averages changed by ~30% over the course of schizogony, with the fastest fork movement and most widely-spaced origins occurring early on.”

Line 102: “The opposite pattern occurs in human cells, where replication speed is usually limited by cellular nucleotide pools and becomes fastest towards the end of S-phase as nucleotide production peaks.”

Results

In the results, it is stated that “…nuclei with the ‘partial’ replication patterns made up 20-30% of the total in P. falciparum at all timepoints, but fewer than 10% of the total in P. knowlesi (Figure 3C)” At the final time point, however, it seems as if the partial replication pattern takes up approximately 30%.

Considering the resolution of the microscopy used in this study and that images were not acquired in 3 dimensions, the authors should consider simplifying the two centrin foci classifications in Figure 4. In 3 dimensions, two centrin foci could be ‘opposite’ to each other but have moved up or down relative to how the viewer sees the image rendered in 2D. Therefore these would appear next to each other but, in reality, be opposite.

The Simon et al. study referenced in the paper quantifies the time P. falciparum nuclei spend with a single or double centrin focus, beginning at about 150 minutes for the first mitosis and continuing at around 100 minutes for subsequent rounds. This observation should be mentioned when discussing the length of time a nucleus has two centrin foci.

The results state that “… nuclei with centrin foci were never seen past the stage of 8-9 nuclei in P. knowlesi (Supp Figure 3).” Despite this, in Supp Figure 3, there are one centrin data points for cells with 10, 11,12, and 15 nuclei.

Treatment of cells with BrdU and EdU induces some DNA damage. For nuclei that have been arrested well before other nuclei in the same cell, the authors should discuss the possibility that BrdU or EdU treatment has caused irreparable double-stranded breaks in these nuclei.

The authors state that the DNA spreading technique used in this study provides a non-uniform stretching factor but then uses a uniform DNA stretching factor determined previously. Could the authors please elaborate on how it is appropriate to apply this stretching factor when the method used provides non-uniform stretching?

Based on the asynchrony in mitosis, it seems improbable that 100% of mononucleated cells would be replicating over 45 minutes, but this is shown in Figure 5D. Is it possible that this could be the product of nucleotide pre-loading? If so, the authors should discuss this.

Figures

Having all the experimental schematics in Figure 1 before the authors explain what they will be used for seems confusing. It may be more intuitive to include each schematic in the figure/panels of the corresponding assay.

It is difficult to get a sense of the spread of the data in graphs 2a,b,e-h; 3d; 5b,d and 6b,d. The authors should consider presenting some of these data as violin plots, which could help to display this spread better.

There is no explanation for what the dotted circle is in Figures 3a and 4b. The authors should detail this in the figure legends.

Based on their use of cyan and yellow for most of their microscopy, it seems the authors are aware of the difficulties of differentiating red, green, and blue for color-blind readers, and this is very much appreciated. However, it should be noted that the yellow and red used to overlay BrdU and EdU are not discernible for multiple types of color-blindness. Therefore, if color-blind friendliness is important to the authors, they should consider changing the red of EdU to magenta or white.

In the results, the authors imply that there were 100 cells analyzed at each time point in Figure 3, but there seems to be far less than 100 data points for many of the time points in Figure 3D. Therefore, the number of cells should be clarified in the figure legend.

In Figures 4B & D, the second row of graphs seemed confusing. The authors should describe these more thoroughly in the figure legend.

Reviewer #3: Minor points

L62 Although cultured P. knowlesi lines can take 28 hours, this is very likely an artifact of culture (in the same way most cultured Pf are in fact shorter than 48h). In this context it would be worth stating it is ~24 hours.

In a separate but related point, whilst the A1-H.1 has previously been noted to have a lifecycle of ~28hr, it appears to be much longer in these experiments. Can the authors mention this at an appropriate point and suggest a reason? I can’t see anything obvious, but it would be interesting to hear the authors thoughts.

L71 P. cynomologi can also be cultured, albeit in macaque cells. Changing this to state cultured in human red blood cells or among human infective species would be more accurate.

L155 Is budding the correct term here? This suggests progeny growing out from a mother cell?

L164 this is an especially interesting finding. Whilst there is clearly no upper limit to the proportion of dividing nuclei, I am not sure it is quite right to suggest that there are not factors limiting it at the individual cell level. It may be that cells that have 100% replicating nuclei in one round, end up dipping down to 20% due to limitations in the next. It would have been really nice to examine this in the context of limiting nucleotide availability, perhaps limiting hypoxanthine or some other precursors.

L188 and Fig 3 - Figure 3 legend could explain B in more detail. Also the Centrin staining is introduced here but not explained until fig 4 and next results section. Authors could look at order in which these are explained to make its inclusion in Fig 3 clearer. I presume that the Pk 33 hour image is a ring? That is probably not obvious for some readers. Define integrated density in the figure legend for 3D.

Fig 4 – probably artifact of compiling but fig 4 has noticeably lower resolution than other figures.

Fig 5 – Although this experiment is really interesting, it took me a really long time to understand the logic involved (probably my failing!). Could a graphic be added to illustrate it or perhaps further explanation? Perhaps adding this to Fig 1 B and C?

L320 convert this to a full sentence rather than parenthetical?

L443 The Pf and Pk data look quite similar, was the trend towards faster movement in Pk skewed by the two very high speed data points, or did it remain if these were removed? Alter description or not accordingly!

PLOS authors have the option to publish the peer review history of their article (what does this mean?). If published, this will include your full peer review and any attached files.

Reviewer #1: No

Reviewer #2: **Yes: **Sabrina Absalon

Reviewer #3: **Yes: **Robert W. Moon
---

## [Decision Letter · Decision Letter 1]

5 May 2022

Dear Catherine,

Thank you very much for submitting your manuscript "DNA replication dynamics during erythrocytic schizogony in the malaria parasites Plasmodium falciparum and Plasmodium knowlesi" for consideration at PLOS Pathogens. As with all papers reviewed by the journal, your manuscript was reviewed by members of the editorial board and by several independent reviewers. The reviewers appreciated the attention to an important topic. Based on the reviews, we are likely to accept this manuscript for publication, providing that you modify the manuscript according to the review recommendations.

As you will see, both reviewers appreciate and thank you for the revisions made to this manuscript. However, reviewer #2 remains concerned about your capacity to accurately enumerate individual parasite nuclei based on wide-field epifluorescence microscopic images. Whilst we consider that the overall thrust of the work is compelling, we agree with the reviewer that caution should be exercised in some of the statements made in the manuscript and that some further alterations are warranted. We therefore request that you consider making the following minor changes in a second round of textual revisions (no additional experimental work is required):

1. With regard to supplementary Figure 2 (Figure S2); we agree that the reviewer makes several well-founded criticisms of your interpretation of these images, but we consider that this figure should be retained in the manuscript in order that the data are available to readers. As requested, please provide information on the microscope and deconvolution system used for the confocal microscopic images in Figure S2 panel B.

2. Please consider making the requested changes to Figure 4, Figure 5 and Figure 6 as requested by the reviewer (Fig 4 B and D: use hour post-invasion instead of nuclei number; Fig 4 C and E, Fig 5 E: use replicating DNA instead of replicating nucleus; Figure 6 B, C, E, and F: use DNA masses instead of nuclei). We would suggest that you make it clear in the figure legends, methods or introductory text that the microscopic methodology used inevitably introduces some uncertainly about actual numbers of nuclei in schizonts.

3. Throughout, as recommended by the reviewer, please use the term schizogony (once defined) in place of ‘cell-division’. Use the term ‘nuclear multiplication’ instead of ‘replication’ when referring to karyokinesis, and use ‘DNA replication’ when referring to chromatin synthesis.

Sincerely,

Michael J Blackman

Associate Editor

PLOS Pathogens

Xin-zhuan Su

Section Editor

PLOS Pathogens

Kasturi Haldar

Editor-in-Chief

PLOS Pathogens

orcid.org/0000-0001-5065-158X

Michael Malim

Editor-in-Chief

PLOS Pathogens

orcid.org/0000-0002-7699-2064

As you will see, both reviewers appreciate and thank you for the revisions made to this manuscript. However, reviewer #2 remains concerned about your capacity to accurately enumerate individual parasite nuclei based on wide-field epifluorescence microscopic images. Whilst we consider that the overall thrust of the work is compelling, we agree with the reviewer that caution should be exercised in some of the statements made in the manuscript and that some further alterations are warranted. We therefore request that you consider making the following minor changes in a second round of textual revisions (no additional experimental work is required):

1. With regard to supplementary Figure 2 (Figure S2); we agree that the reviewer makes several well-founded criticisms of your interpretation of these images, but we consider that this figure should be retained in the manuscript in order that the data are available to readers. As requested, please provide information on the microscope and deconvolution system used for the confocal microscopic images in Figure S2 panel B.

2. Please consider making the requested changes to Figure 4, Figure 5 and Figure 6 as requested by the reviewer (Fig 4 B and D: use hour post-invasion instead of nuclei number; Fig 4 C and E, Fig 5 E: use replicating DNA instead of replicating nucleus; Figure 6 B, C, E, and F: use DNA masses instead of nuclei). We would suggest that you make it clear in the figure legends, methods or introductory text that the microscopic methodology used inevitably introduces some uncertainly about actual numbers of nuclei in schizonts.

3. Throughout, as recommended by the reviewer, please use the term schizogony (once defined) in place of ‘cell-division’. Use the term ‘nuclear multiplication’ instead of ‘replication’ when referring to karyokinesis, and use ‘DNA replication’ when referring to chromatin synthesis.

Reviewer Comments (if any, and for reference):

Reviewer's Responses to Questions

**Part I - Summary**

Reviewer #1: In light of the comments made by the other reviewers and despite my initial assessment I’m happy to trust judgement of the editor about the general suitability of this manuscript for publication in Plos Pathogens. Concerning the scientific integrity of the manuscript all my comments have been sufficiently addressed or at least contextualized appropriately. Hence, I have no further reservations recommending acceptance of this manuscript as is and want to congratulate the authors on their study, which offers an interesting complementary view on the process of schizogony.

Reviewer #2: I want to thank the authors for generating a new set of biological replicates, which strengthens the robustness of the findings. I also want to thank the authors for reaching out to Drs. Ganter and Guizetti best represent their data in the current manuscript and include the most recent published data in the discussion section. However, in the absence of a fluorescent nuclear envelope marker, epifluorescence microscopy does not provide the required resolution to resolve nuclei in multinucleated schizont parasites. In conclusion, I have raised significant comments that need to be addressed in the data representation

in figures 4, 5, and 6 upon acceptance for publication.

**Part II – Major Issues: Key Experiments Required for Acceptance**

Reviewer #1: n.a.

Reviewer #2: My last and significant disagreement with the authors remains on the ability to distinguish and resolve a single nucleus in multinucleated schizont parasites without a fluorescent nuclear envelope marker and epifluorescence microscopy.

I am pleased the authors referred to one of my favorite FIBSEM papers by Rudlaff et al. The introduction section explains the limitation of standard fluorescent microscopy.

"The attainable resolution of standard fluorescent microscopy is defined by the Abbe diffraction equation (resolution = λ / (2 * NA), where λ is the wavelength of light used–typically 350–700 nm–and NA is the numerical aperture of the objective–typically <1.5). Practically, this limits the amount of information gained from immunofluorescence microscopy to a resolution of ~200 nm, which does not generally allow separation of structures smaller than this size."

In this paper, the authors mentioned that to generate a robust rendering of a single nucleus, and they called "Nuclear connections for 4n and 2n nuclei were at least 100nm wide through at least 5 Z-sections (100nm deep) to ensure that the observations of connected nuclei were robust" see Data analysis section.

In epifluorescence microscopy, the z-resolution is 2 to 3 times the XY resolution. So therefore, even if the resolution is at 200 nm XY, the z resolution would be between 400 nm to 600 nm, higher than the width of a single nucleus, which ranges from 200 to 350 nm.

Therefore epifluorescence images using DNA staining without a fluorescent nuclear envelope marker can reveal an estimate of DNA masses but not individual nuclei.

Regarding support figure 2, I have multiple concerns, and it supports the fact that only DNA masses and not individual nuclei can be visualized.

For instance, in Panel A, parasite 1: the two -EdU masses with the individual 1C could be either an individual nucleus or a single nucleus at the mitotic spindle phase, parasite 2, the two +EdU top left look very much like one nucleus at the anaphase stage upon nuclear fission, or they could be the individual nucleus. Again the current experimental setting allows to call for DNA masses or replicating DNA and not for individual nucleus.

The authors compared individual z-stack with the maximum projection obtained by wide-field microscopy based on panel B. Confocal microscopy relies on using a pinhole to collect only the fluorescence at the focal point resulting in increasing the resolution. In other terms, with confocal imaging, the fluorescence from objects outside the focal plane is not detected outside of the focal plane, which is not the case in the figure represented in panel B. Could the authors elaborate on what microscope and deconvolution program they use to run their confocal imaging?

In addition, as the author mentions in their manuscript (discussion section): " the centrin foci clearly disappeared in late schizonts after the final round of karyokinesis" therefore, the authors must agree that the third parasite in supplementary figure 2 panel B represents unspecific staining with anti-centrin antibody.

I recommend removing supplementary figure 2 for the final manuscript.

In conclusion, the authors must make the following modifications to their figures and manuscript, ensuring a robust conclusion of their data without changing the main finding of the manuscript.

• Figure 4 Panel B and D: use hour post-invasion instead of nuclei number

• Figure 4 Panel C and E, figure 5 Panel E: use replicating DNA instead of replicating nucleus

• Figure 6 Panel B, C, E, and F: us DNA masses instead of nuclei

**Part III – Minor Issues: Editorial and Data Presentation Modifications**

Reviewer #1: n.a.

Reviewer #2: The terms cell cycle and schizogony have been used interchangeably in the manuscript (for instance, in the discussion section: lines 370, 371). I would recommend the authors to use the term cell division or stick to schizogony not to confuse a non-malaria biologist. In the same line, it would be best to use the term nuclear multiplication instead of replication when the authors refer to karyokinesis and DNA replication when they refer to chromatin synthesis.

PLOS authors have the option to publish the peer review history of their article (what does this mean?). If published, this will include your full peer review and any attached files.

Reviewer #1: No

Reviewer #2: **Yes: **Sabrina Absalon

Figure Files:

Data Requirements:

Reproducibility:

References:

---

## [Editor Report · Decision Letter 2]

13 May 2022

Dear Catherine,

We are pleased to inform you that your manuscript 'DNA replication dynamics during erythrocytic schizogony in the malaria parasites Plasmodium falciparum and Plasmodium knowlesi' has been provisionally accepted for publication in PLOS Pathogens.

Best regards,

Michael J Blackman

Associate Editor

PLOS Pathogens

Xin-zhuan Su

Section Editor

PLOS Pathogens

Kasturi Haldar

Editor-in-Chief

PLOS Pathogens

orcid.org/0000-0001-5065-158X

Michael Malim

Editor-in-Chief

PLOS Pathogens

orcid.org/0000-0002-7699-2064
---

## [Editor Report · Acceptance letter]

11 Jun 2022

Dear Dr Merrick,

We are delighted to inform you that your manuscript, "DNA replication dynamics during erythrocytic schizogony in the malaria parasites *Plasmodium falciparum* and *Plasmodium knowlesi*," has been formally accepted for publication in PLOS Pathogens.

Best regards,

Kasturi Haldar

Editor-in-Chief

PLOS Pathogens

orcid.org/0000-0001-5065-158X

Michael Malim

Editor-in-Chief

PLOS Pathogens

orcid.org/0000-0002-7699-2064